# Wind enhances differential air advection in surface snow at sub-meter scales

Stephen A. Drake[1], John S. Selker[2], Chad W. Higgins[2]

[1]College of Earth, Ocean and Atmospheric Sciences, Oregon State University, Corvallis, 97333, USA
[2]Biological and Ecological Engineering, Oregon State University, Corvallis, 97333, USA

*Correspondence to*: Stephen A. Drake (stephenadrake@gmail.com)

**Abstract.** Atmospheric pressure gradients and pressure fluctuations drive within-snow air movement that enhances gas mobility through interstitial pore space. The magnitude of this enhancement in relation to snow microstructure properties cannot be well predicted with current methods. In a set of field experiments we injected a dilute mixture of 1% carbon
monoxide and nitrogen gas of known volume into the topmost layer of a snowpack and, using a distributed array of thin film sensors, measured plume evolution as a function of wind forcing. We found enhanced dispersion in the streamwise direction and also along low resistance pathways in the presence of wind. These results suggest that atmospheric constituents contained in snow can be anisotropically mixed depending on the wind environment and snow structure, having implications for surface snow reaction rates and interpretation of firn and ice cores.

**1 Introduction**

Atmospheric pressure changes over a broad range of temporal and spatial scales stimulate air movement in near-surface snow pore space (Clarke et al., 1987) that redistribute radiatively and chemically active trace species (Waddington et al., 1996) such as $O_3$ (Albert et al., 2002), OH (Domine and Shepson, 2002) and NO (Pinzer et al., 2010) thereby influencing their reaction rates. Pore space in snow is saturated within a few millimeters of depth in the snowpack that does not have
large air spaces (Neumann et al., 2009) so atmospheric pressure changes induce interstitial air movement that enhances snow metamorphism (Calonne et al., 2015; Ebner et al., 2015) and augments vapor exchange between the snowpack and atmosphere. The relative influences of different pressure-driven processes in the near surface snowpack are not well understood but are important to distinguish because different processes disperse water vapor and trace species with different signatures. Pressure-driven air movement in snow pore space has been generally termed "pressure-pumping" (Massmann
and Frank, 2006) or more specifically "wind-pumping" (Colbeck, 1989; Clarke and Waddington, 1991; Albert et al., 2002) in circumstances where localized wind characteristics strongly impact variability in the pressure field. To clarify the importance of discriminating the influence of advection relative to diffusion in snow pore space, we briefly describe dominant in-snow processes forced by atmospheric pressure changes.

In his foundational publication on the topic of wind-pumping, Colbeck (1989) described how atmospheric pressure changes drive a variety of in-snow processes. Towards the low frequency end of the pressure change continuum, synoptic weather evolution imparts approximately hydrostatic surface pressure changes over multi-day time spans (Wallace and Hobbs, 2006). As a synoptic-scale high-pressure system builds over a given site, compression of the air column pushes air that was formerly just above the snowpack into the snowpack. As the high-pressure system weakens over the site, the air column expands and air that formerly was contained in the near-surface snow pore space translates into the atmosphere. Each high/low pressure transition corresponds to a single "breath" by the snowpack that vertically exchanges air contained in the near-surface pore space of the snowpack with air just above it. Although the amplitude of synoptic pressure changes is large (~4000 Pa) relative to turbulently-generated pressure changes (< 100 Pa), (Paw U et al., 2004; Drake et al., 2016), they are infrequent and therefore cause negligible snow/atmosphere fluxes (Bartlett and Lehning, 2011). Synoptic pressure changes have been linked to mixing in firn, however, (Severinghaus et al., 2010) and may also leave an isotopic signature of synoptic intensity in ice cores (Buizert and Severinghaus, 2016). Filtering of fine particle atmospheric constituents by snow (Waddington et al., 1996) at synoptic frequencies would correspondingly be gradual and relevant over seasonal and longer time scales.

By contrast, localized pressure changes due to wind blowing over surface features or caused by wind variability (turbulence) generate pressure changes with much higher frequency, smaller spatial extent and smaller amplitude than synoptic-scale pressure changes. Wind blowing steadily over surface features generates localized, quasi-static pressure gradients and air in snow moves in response to these wind-induced pressure gradients (Colbeck, 1989). These topographically induced pressure gradients generate quasi-stationary circulation patterns that transport gases (Massman and Frank, 2006) and form zones of preferential sublimation and deposition (Albert, 2002) and therefore have a discernable advective signature. Turbulently generated pressure fluctuations induce air movement in snow (Drake et al., 2016) but the response of air contained in snow pore space to turbulent forcing above the snow or above permeable media in general is not fully understood despite considerable effort (de Lemos et al., 2006; Mößner and Radespiel, 2015; many others). Classical boundary layer theory (Beavers and Joseph, 1967) suggests that the time-averaged pressure gradient in permeable media such as snow would generate Darcian flow (advection) aligned with the pressure gradient. Unlike the advection signature for topographic forcing, the turbulence signature does not exhibit quasi-static circulation patterns. Similar to turbulence, advection through a mechanically dispersive medium such as snow dissipates a concentration gradient (Scheidegger, 1954) but in this case preferentially spreads a plume more aggressively in the downstream direction.

Airflow through snow is regulated by intrinsic permeability, which is a proportionality constant in Darcy's Law and is a measure of the interconnectedness of the pore space. Snow permeability is difficult to measure in field conditions but is a fundamental input parameter to model in-snow advection (Darcian flow). Currently accepted sampling techniques to obtain snow permeability include both direct measurements and indirect measurements that infer permeability from some other snow property such as specific surface area and/or snow density. Sub-liter sized snow samples are typically used to obtain direct permeability measurements with a flow-through permeameter (Courville et al., 2007), microtomography (Calonne et

al., 2012) and when using an integrating sphere to obtain SSA measurements and infer permeability (Gallet et al., 2009). A near-infrared photography technique that infers SSA from reflectance (Painter et al., 2007; Tape et al., 2010) acquires pore space characteristics over larger areas but only in two dimensions, as do stereological measurements (Davis et al., 1987; Matzl and Schneebeli, 2010) for smaller sample sizes. Active acoustic techniques of inferring large-footprint, volume-averaged permeability of snow cover have shown potential (Albert, 2001; Drake et al., 2017) but these techniques are unproven for standard data collection. Methods that correlate SSA and snow density with intrinsic permeability offer promise to relate small sample sizes to larger snow volumes. However, none of these standard techniques sample intrinsic permeability of large snow volumes and therefore they do not capture macroscopic changes in permeability due to snow inhomogeneities and fractures. The consequence of neglecting the variability of intrinsic permeability for modelling airflow through snow is not known.

The presence of in-snow advection has been experimentally inferred from natural convection (Sturm and Johnson, 1991) and from temperature changes caused by forced ventilation Albert and Hardy (1995), Sokratov and Sato (2000) and from $CO_2$ flux measurements (Bowling and Massman, 2011) but few measurements of natural air advection in snow have been obtained (Albert and Shultz, 2002; Huwald et al., 2012). Bulk $CO_2$ flux measurements by Massmann and Frank (2006), Seok et al. (2009), and Bowling and Massman (2011) have increased our appreciation for the role of wind-pumping in enhancing soil/snow/atmosphere exchange beyond that given by diffusion but lack the spatial and temporal granularity needed to discern between the relative roles of in-snow transport processes. A deeper understanding of the processes that link atmospheric pressure forcing to in-snow pore space response is needed if we are to accurately model how water vapor and chemically and radiatively active trace species propagate into, through, and out of the snow pore space.

The overarching goal of this experiment is to measure wind forcing above the snow and simultaneously perform high-spatial and temporal measurements of the evolution of a trace gas release in snow such that we can link wind forcing with in-snow response. Our strategy is to compare model simulations that implement a solution of the advection/diffusion equation for homogenous, permeable media with experimental measurements of dispersion of a tracer gas in snow. Anisotropy of seasonal snow has been evaluated (Calonne et al., 2012) and we do not assume snow homogeneity in our experimental design. Rather, we compare field experiments with an analytical solution for dispersion in homogenous media to highlight the influence of snow inhomogeneities. Step changes in permeability between successive snow layers further complicate the relationship between wind forcing and the in-snow advective response (Colbeck, 1991; Albert, 1996). To minimize the complicating influence of snow layering, we confined this exploration to the topmost snow layer that had been deposited by a significant snowfall event. We therefore focus this investigation on the effect that wind blowing over snow has on air movement within the topmost layer of a snowpack.

## 2 Methods

### 2.1 Snow picket description

The measurement network was composed of 28 thin-film Applied Sensor MLC carbon monoxide (CO) sensors with detection range spanning four orders of magnitude (from 0.5 to 500 ppmv). To measure CO gas at well-known positions in the snow, seven CO sensors were mounted in 15-cm intervals on each of four tapered poles (or snow pickets, Fig 1) with dimensions 1 m x 5 cm. We deployed the snow pickets in the topmost snow layer that was at least 20 cm deep to minimize the influence of the profile of the instruments in disrupting interstitial flow. The pickets were inserted horizontally into the snow forming a rectangular sensor grid. Silicone tubing strung down the center of each picket to an outlet opposite the CO sensors provided a means to deliver the CO gas to a well-known position in the snow. This same system was also used in Huwald et al. (2012), and we refer readers to that document for a thorough explanation of materials, manufacturing and wiring requirements.

Data from the 28 CO sensors were acquired on two synchronized Campbell Scientific CR1000 loggers at 1-minute intervals. CO gas was delivered fast enough relative to the 1-minute measurement interval to approximate an instantaneous release. A Campbell Scientific Irgason ultrasonic anemometer captured 3-D wind components at 20 Hz approximately 1 m above the snow. One-minute wind speed and direction averages were computed by post-processing the high-frequency data. The experiment configuration is presented as a schematic diagram in Fig. 2.

### 2.2 Tracer gas

Consistent with Huwald et al. (2012), we chose CO as a tracer gas because its molecular weight is very close to that of air so it is nearly neutrally buoyant. Neutral buoyancy ensures that gravitational effects do not influence plume evolution, although neutral buoyancy is not strictly achieved for this experiment. In practice, neutral buoyancy is difficult to achieve because the air space in snow is saturated and therefore more buoyant than dry air but less buoyant than $N_2$. CO can be safely handled when used in small quantities, has low background concentration and low water solubility. This latter consideration is important because snow grains may be coated with a thin film of liquid water, even at sub-freezing temperatures (Boxe and Saiz-Lopez, 2009). A mixture of 1% CO in 99% $N_2$ provided sufficient concentration for sensor detection.

### 2.3 Site description

The system represented in Fig. 2 was deployed at three locations: Santiam Pass, Oregon (elevation: 1468 m); Dutchman Flat Sno-Park, Oregon (elevation: 1905 m); and Storm Peak Lab (SPL), Mt. Werner, Colorado (elevation: 3220 m) during winter and spring seasons of 2014 and 2015, respectively. These sites span a broad range of wind forcing and snow density from 227 kg-m$^3$ to 430 kg-m$^3$. The Santiam Pass and Dutchman Flat sites were nearly flat, while the SPL deployment site was located on a gradual slope (~ 17% average slope over 100 m distance, as determined from Google Earth) with a western

aspect. For each deployment we mounted a sonic anemometer at ~ 1-m height on a low-profile tower with the sonic transducer facing into the prevailing wind. The windward side of the tower was kept free of disturbance.

## 2.4 Data selection criteria

Data were selected from a larger data set composed of 24 releases in 10 different snow conditions over two field seasons. Quality control criteria excluding some data were weather-related instrument issues such as sonic anemometer transmission losses due to snow/rain/riming, excessive gas leakage around the release picket, gas leakage at a tube fitting, and excessive icing/wetting/temperature changes of CO sensors. The dual requirements that the CO sensors needed to be ice-free with minimal temperature variations for optimal operation restricted deployment time spans to no more than several hours. Between deployments the sensors were air-dried in a lab environment to return them to an optimal operational state. Immediately prior to each deployment, we determined the prevailing wind direction so that the pickets could be inserted approximately perpendicular to the wind direction. This orientation maximized the sensor network's ability to resolve a plume propagating downwind and was achieved for all cases except cases 13 and 14, which experienced a wind shift during instrument setup. With the exception of case 12 the surface snow layer was deposited by a single storm event and was sufficiently deep to position the sensor pickets in a discrete snow layer. Cases 7 through 11 were acquired in a snow layer that had undergone equilibrium snow metamorphism forming spheroidal snow grains. Snow density was acquired with a 1000cc Snowmetrics RIP 1 Cutter and weighed with a digital gram scale.

## 2.5 Calibration

We built a calibration chamber (Fig. 3) into which we could simultaneously place all 4 pickets and verify the operational capabilities of the Applied Sensor MLC CO sensors that we used for this experiment. We found through trial and error that the CO sensors are sensitive to both humidity and temperature. Furthermore, we found that sensor sensitivity decreases when exposed to the same CO concentration over a long period (> 10 minutes). To overcome these deficiencies we transported the calibration chamber to the Santiam Pass warm snow site where the surface now layer was -1 ºC and the Storm Peak Lab cold snow site where the surface layer temperature averaged -6 ºC during the calibration time period. The procedure was to shovel some snow into the chamber then insert the pickets into support collars that suspended the pickets in the chamber above the snow. The chamber was sealed and a measured volume of CO introduced. A fan inside the calibration chamber facilitated thoroughly mixing the air. Measurements were acquired at 1-minute intervals with two Campbell Scientific CR1000 loggers until sensor response was documented. The chamber was then opened on both ends and allowed ~ 30 minutes for the chamber to fully evacuate. This procedure was repeated for the next measurement. This calibration procedure was time consuming and therefore we were not able to perform it in concert with field deployments. Although the maximum recommended operating concentration is 500 ppmv, tests revealed a linear response that consistently exceeded 1000 ppmv. Measured gas concentration at the sensor nearest the release point typically exceeded 1000 ppmv but this sensor was not

used for analysis and therefore did not influence experimental results. Further calibration details can be found in Huwald et al. (2012).

## 3 Data analysis

### 3.1 Methodology

We used two methods to analyze the results. For the first method, we applied calibration coefficients from either warm or cold snow calibrations to voltage measurements and derived CO concentration for each sensor at each time step. From these data we calculated the position of the center of mass in order to determine plume propagation speed. In the second method, we calculated the time required to reach maximum concentration at each sensor as a measure of plume propagation. Deviations from the simulated concentration give a measure of the influences of advection and snow heterogeneity.

A solution of the 3-D advection/diffusion equation for a point source (with 2-D advection) is (Socolofsky and Jirka, 2005):

$$C(x, y, z, t) = \frac{M}{4\pi t \sqrt{4\pi t D_x D_y D_z}} \exp\left[ -\frac{((x-x_1)-Ut)^2}{4tD_x} - \frac{((y-y_1)-Vt)^2}{4tD_y} - \frac{(z-z_1)^2}{4tD_z} \right], \tag{1}$$

where $C(x, y, z, t)$ is concentration, $M$ is mass, $U$ and $V$ are horizontal wind components, $D_{x,y,z}$ are diffusivity along the x, y and z axes, and $t$ is time. For idealized homogenous snow (constant $D_{x,y,z}$) we can differentiate Eq. (1) with respect to time and set this result equal to zero to find the streamwise advection velocity:

$$\sqrt{U^2 + V^2} = \sqrt{\frac{r^2 - 6Dt_{MAX}}{t_{MAX}^2}}, \tag{2}$$

where $r$ is the distance from the release point to a given position and $t_{MAX}$ is the time interval between the release time and the time at which maximum concentration is reached at that position. For zero wind velocity:

$$D = \frac{r^2}{6t_{MAX}}. \tag{3}$$

Equation (3) permits calculation of the horizontal diffusivity as a function of the time required between the initial release and
the maximum measured concentration at each sensor location, assuming zero air velocity in snow. Non-zero interstitial air velocity and snow heterogeneity manifest as spatial variations in diffusivity. For each CO release, we measured CO concentration as a function of time and distance from the release point to find $t_{MAX}$ for each sensor. Using Eq. (3) we then calculated the diffusivity for each sensor and subtracted these values from a mono-valued diffusivity of 2.56 x $10^{-5}$ m$^2$ s$^{-1}$ consistent with snow (Huwald et al., 2012) to derive a residual that is an approximation of wind-driven dispersion
enhancement. This technique has the advantage that absolute concentration is irrelevant so the result is insensitive to sensor calibration error.

**3.2 Caveat**

We will not attempt to compare the vertical diffusivity with the horizontal diffusivity, as that comparison would require a 3-dimensional measurement network. Calonne et al. (2012) found that snow anisotropy causes differences between horizontal and diffusivity for air in snow. We postulate that in-snow vertical transport enhancement increases as wind ventilation

increases (Albert and Shultz, 2002). For this reason, we expect that our computations using Equation (3) will be systematically low, the degree to which depends on the difference between the horizontal and vertical diffusivity and the relative wind enhancement of in-snow air motion in the horizontal and vertical directions. As our measurements will show, a distinct snow layer is not homogenous and this method will highlight macroscopic channels of inhomogeneity.

**4 Results**

**4.1 Model simulations**

Results in a horizontal plane from an advection/diffusion model assuming isotropic media with a diffusivity of $2.56 \times 10^{-5}$ $m^2$ $s^{-1}$ are shown in Fig. 4a. An instantaneous release at the origin (marked by a red asterisk) spreads in time and the red dots mark the locations of point measurements at 15 cm, 30 cm and 45 cm from the release point. Half-hour time series of the simulated concentrations at these three locations are shown in Fig. 4b. The three time series in Fig. 4b delineate

"breakthrough curves" that share a distinctive shape with a rapidly rising concentration to a peak followed by gradual decay. In a purely diffusive environment the breakthrough curve of each successively distant point from the release location is contained within curves defined by closer points as in Fig. 4b. Isotropic molecular diffusion spreads a plume in all directions but the centroid of mass remains stationary over time. On the other hand, advection translates the centroid. In Figs. 4c and 4d we have imposed an advective flow of 0.5 mm $s^{-1}$ oriented along the positive x-axis. For the corresponding set of

breakthrough curves in Fig. 4d, the traces cross at some point in time if advection is sufficiently fast relative to diffusion. By comparing breakthrough curves derived from field experiments with simulations we can estimate the relative influences of dispersive processes. In this idealized description we do not account for snow heterogeneity, which enhances diffusion in regions of higher porosity, potentially leading to centroid displacement.

**4.2 Field data**

The results presented in this paper are based on 14 CO releases selected from five snow conditions. Distance between snow pickets, picket depths, release volumes and associated weather conditions and surface layer snow metrics are detailed in Table 1. With the exception of case 12, pickets were placed in a layer of snow generated by a single snow event such that we could minimize the effect of snow layering on dispersion. Release volume was measured with an Aalborg GFM17 Mass Flowmeter (range 0-15 SLPM) for cases 1 through 11 and with a Precision Sample Magnum Series 500 ml gas tight syringe

for cases 12 through 14. For case 12, the pickets were placed below an ice layer that was overlain by surface snow generated

by a single storm event. Cases 1-11 were calibrated using a warm snow calibration at Santiam Pass and cases 12-14 were calibrated with a cold snow calibration at Storm Peak Lab. At each site we carefully dug a shallow trench, exposing a clean face of the snow layer into which we inserted the sensor-mounted pickets. Once the pickets were placed, we backfilled the snowpit with fresh snow and smoothed the surface to match the surrounding, undisturbed snow level. Snow in the backfilled

trench was not measured directly, however, it is possible that disturbing snow in close proximity to the measurement volume could have influence the local pressure field. In Huwald et al. (2012) sensor pickets were placed vertically and the authors noted leakage around the picket perimeter that manifested as enhanced dispersion along the picket axis for low snow density releases. Even with horizontal (and buried) picket placements we observed indication of leakage for a few cases but most cases were performed in snow that seated snugly against the pickets, minimizing along-picket leakage.

### 4.3 Breakthrough curves

A picture of the experiment setup for March 26, 2015 is shown in Fig. 5 and the results of CO release R2 are plotted in Fig 6 (case 14 in Table 1) in the presence of 3.06 m-s$^{-1}$ mean wind speed and 227 kg-m$^{-3}$ density snow. The breakthrough curves are smooth and indicative of diffusion-dominated dispersion yet a subtle advection signature of breakthrough curves crossing

each other is evident in Fig. 6b, similar to Fig. 4d. This result shows that an advection signature is evident at approximately 20 cm depth in mid-to-low density snow. The maximum concentration measured at the release point in Fig. 6a exceeded the linear calibration range of the CO sensors. Calibration range exceedance near the release point commonly occurred during releases but the results presented in this paper do not rely upon release point concentration measurements. Rather, larger releases enabled greater resolution of far field concentration measurements and $t_{MAX}$ calculations.

### 4.4 Effect of wind direction

For two contrasting cases during the same deployment (April 19[th], 2014), Figs. 7a-d shows the effect of wind on subsurface plume evolution. In case 8 (Figs. 7a-b), prevailing winds were persistent and from the west whereas for case 11 (Figs. 7c-d) winds were light and variable. The time required to reach the maximum concentration for each sensor as described in Eq. (3) is plotted in Figs. 7b and 7d. For case 8 (Fig. 7b) the plume orients streamwise to the wind with increased streamwise

dispersion relative to cross-stream. The plume shape in low wind case 11 (Fig. 7d) is more circular as would be expected for diffusion-dominated dispersion. We attribute deviations from radial symmetry for low-wind case 11 to snow inhomogeneities. This case comparison shows that an advective signature is evident in at 9 cm depth in dense (430 kg m$^{-3}$) snow.

The subtle, streamwise plume alignment evident in Fig. 7a-b is more easily discriminated with larger CO releases, an

example of which is plotted in Fig. 7e-f. This release was too large to approximate a point release but it unambiguously shows that the plume aligns in a streamwise orientation. Preferential streamwise dispersion was duplicated for subsequent large releases with persistent prevailing winds (cases 8 and 9, not shown). In-snow, streamwise plume alignment under a

persistent wind regime is an unsurprising result that nevertheless bears reporting because previous studies have lacked a sufficiently dense sensor network to resolve it. Given the Clifton et al., 2008 result that air in permeable media pore space responds to a pressure gradient rather than shear, a persistent in-snow advective flow indicates a persistent in-snow pressure gradient.

**4.5 Plume propagation given by centroid of mass**

We calculated the displacement of the centroid of mass at each time step as an indicator of advection. The centroid of mass is a more stable measure of plume propagation than the maximum concentration location because the latter may vacillate at each time step when two measurements are nearly the same. In Fig. 8a we plotted the position of plume centroid relative to release position for case 13, color-coded by minute since release. The black asterisk marks the release point. Circles delineate centroid position at each minute and the circle diameter is a relative measure of the plume mass-weighted RMS distance from the center of mass. Triangles in Fig. 8a indicate anticipated one-minute translation given by wind direction at 1/1000 of the wind speed. This multiplicative factor (1/1000) reduces wind speed to on the order of mm $s^{-1}$, the approximate order of magnitude for air advection in snow (Huwald et al., 2012). We do not account for the mass that advects out of the measurement network because we lack 3-dimensional measurements needed to simultaneously constrain mass loss in the vertical and horizontal directions. Instead, we assume that the centroid position in the horizontal plane is accurate over a short timespan between the initial release and the time at which mass starts to advect out of the perimeter of the measurement network.

For the initial several minutes after the release, the calculated centroid position was indeterminate because the sensor at the release position had saturated, returning NaN (Not a Number) results. After several minutes more sensors detected the plume so the centroid position stabilized as it propagated downwind. While propagating downwind some of the plume mass concurrently propagated vertically, out of the sensor network plane. We anticipate that horizontal diffusion was slowed to the degree that the vertical diffusivity exceeded the horizontal diffusivity and the center of mass of the buoyant plume lifted. Numerical simulations similar to those shown in Fig. 4 but using a vertical diffusivity that decreases with snow depth (not shown) are consistent with this hypothesis. After approximately 13 minutes, the calculated centroid position was driven by the CO mass still in the horizontal plane and within the sensor network and the centroid of mass appeared to stall because the smaller horizontal footprint of the upward moving plume. For the time span between 3 min and 13 min, the center of mass advected 6 cm giving an average velocity of $1.0 \times 10^{-4}$ m $s^{-1}$, slightly lower than $1.2 \times 10^{-4}$ m $s^{-1}$ reported by Huwald et al. (2012) for higher density snow (360 kg-$m^{-3}$). We have no measure of specific surface area, which we acknowledge influences permeability (Calonne et al., 2012) and acknowledge imprecision in the center of mass advection speed determination.

To further assess the influence of wind on plume propagation we calculated the angle between the propagation of the center of mass and that given by wind direction, again assuming the wind-driven advection speed was ~1/1000 of the wind speed. Results for 4 representative cases are shown in Fig. 8b. In low-density snow and moderate wind speeds (case 13, 227 kg-$m^{-3}$

snow density, 3.41 m-s$^{-1}$ wind speed), wind direction was a good predictor for plume propagation direction, even at approximately 20 cm depth. In denser snow and moderate wind speeds (case 8, 430 kg-m$^{-3}$ snow density, 2.19 m-s$^{-1}$ wind speed) wind direction remained a good predictor for plume propagation direction with the exceptions that several minutes were required for the plume centroid position to stabilize (as noted in Fig 8a) and after ~ 13 minutes by which time snow heterogeneity and vertical dispersion degraded the correlation between wind direction and center of mass propagation direction. For case 11, in which snow density was high and winds were light and variable (430 kg-m$^{-3}$ snow density, 0.38 m-s$^{-1}$ wind speed), the angle between wind direction and plume propagation as indicated by the center of mass was larger and highly variable. For case 12, in which low-density snow overlaid an ice layer (variable snow density, 4.05 m-s$^{-1}$ wind speed), the plume diverged into several preferred pathways (see Section 4.6) but the centroid of mass generally advected downwind. These results show that the centroid of mass propagates downwind more reliably in lower density snow than in dense snow but the centroid of mass propagation direction is not necessarily a reliable indicator of the evolution of the plume footprint.

### 4.6 Snow layering

With the exception of case 12, we deployed our equipment in the topmost layer of a significant snowfall event to minimize the influences of snow layering on plume evolution. We include case 12 because an 18 cm new snowfall event overlaid a thick (~5-10 mm) and pervasive layer of refrozen ice, providing an unusual opportunity to study the dispersion signature below a buried, ~ impervious layer where one might expect no advection. The CO sensor-mounted snow pickets were placed at roughly 18 cm depth, just below the ice layer. Surface snow was unevenly distributed over the ice layer so picket depths varied by up to 5 cm but all pickets were positioned at the same depth below the ice layer. Winds were persistently strong from the west (blowing left to right in Fig. 9) during this release. Plume evolution for this case indicates that dispersion had not only molecular diffusion and directional wind signatures but was also characterized by preferential flow regions. We hypothesize that the tracer gas was preferentially following incipient cracks and more porous pathways in the snow layer. This hypothesis is supported by Nachshon et al. (2012), who found that fractures in soil increase permeability by several orders of magnitude and serve as preferred flow pathways in the presence of a mean background flow. From this result we conclude that pressure changes above the snow incite air movement through incipient cracks and porous zones below low permeability ice layers but with less wind-directional influence than that seen for a surface snow layer.

### 4.7 Plume propagation given by time to maximum concentration

Given the previously mentioned deficiencies with computing dispersion enhancement from centroid of mass velocity, we alternatively investigate dispersion enhancement by comparing measurements with the result given by Eq. (3) using a molecular diffusivity of CO in snow, $(D_{CO})$, of 2.56 x 10$^{-5}$ m$^2$ s$^{-1}$ (from Huwald et al., 2012). We note that diffusivity of a gas in snow varies with temperature, pressure and snow state. However, these parameters do not vary significantly for different cases obtained during a given experiment deployment. We use the diffusivity, $D(t_{MAX})$, calculated from Eq. (3) and $D_{CO}$ to estimate the Péclet number:

$$Pe = \frac{advective\ transport\ rate}{diffusive\ transport\ rate} = \frac{D(t_{MAX}) - D_{CO}}{D_{CO}}. \tag{4}$$

In Equation (4), we note that measured $D(t_{MAX})$ includes influences of both molecular diffusion and advection so one must subtract the diffusive component ($D_{CO}$), from the measured $D(t_{MAX})$ to derive the advective component. Since the plume is preferentially spreading vertically, mass is lost from the horizontally oriented measurement network, systematically

increasing $t_{MAX}$ values and thereby decreasing measured effective diffusivity to values smaller than molecular diffusivity. To compensate for the systematic depression of measured effective diffusivity, we find the difference between the smallest calculated effective diffusivity and the molecular diffusivity and normalize measured effective diffusivity by this difference:

$$Pe_{norm} = \frac{D(t_{MAX}) + [D_{CO} - MIN(D(t_{MAX}))] - D_{CO}}{D_{CO}} = \frac{D(t_{MAX}) - MIN(D(t_{MAX}))}{D_{CO}}. \tag{5}$$

Our rationale for applying this normalization is that the dispersion of a gas in snow can be no less than the molecular

diffusivity. This normalized Péclet number no longer represents the absolute ratio of advective to diffusive transport. But it does preserve a relative measure of advection vs. diffusion and guarantees that the normalized Péclet number is no less than zero. For example, comparing the normalized Péclet number for moderate-wind case 8 (2.19 m-s$^{-1}$ wind speed) in Fig. 10a (see also Figs. 7a-b) with low wind case 11 (0.38 m-s$^{-1}$ wind speed) in Fig 10b (see also Figs. 7c-d) we note advective signatures streamwise and also along one of the sensor pickets, indicating that leakage along a picket that is exacerbated by a

wind-induced pressure gradient. The bullseye at the sensor just below the release point in Fig. 10b is an artifact of the large release volume rather than advection. The $Pe_{norm}$ gradient is weak in Fig. 10b relative to Fig. 10a as one would expect for a diffusion-dominated regime. Discrepancies from radial symmetry evident in Fig. 10b indicate preferential dispersion pathways along inhomogeneities in the snow layer. Ignoring differences in sensor depth, snow microphysics, and high volume release cases (cases 4, 5, 6 and 10) the $R^2$ correlation was 0.61 between average wind speed and the maximum

$Pe_{norm}$ for the remaining cases. This result suggests that in-snow advection increases with wind speed and that snow state and depth in snow tempers the magnitude of in-snow advection in a given layer. A three-dimensional measurement design would improve the quality of the Péclet number values and, accompanied by high-resolution snow characterization, enable absolute comparison of the advective vs. diffusive transport in both vertical and horizontal planes.

## 5 Conclusions

Atmospheric pressure gradients can induce subsurface advection that enhances plume dispersion, even in dense snow. Beneath an ice layer the evolution of a tracer gas plume indicates signatures of enhanced diffusion and advection along high permeability pathways in the presence of wind. Over smooth, flat reaches with a prevailing wind, a subsurface plume aligns in a streamwise orientation. Snow inhomogeneities can enhance anisotropic dispersion as wind speed increases, challenging the notion of using small sample size to represent the intrinsic permeability of snow over broad regions. By comparison of a

normalized Péclet number between cases with different wind forcing, we find that variability in the advection signature increases with wind speed.

We were not able to discriminate the relative importance of different processes that enhance in-snow air movement with a 2-D configuration of the sensor network. We anticipate that a modified 3-D deployment design that has a smaller instrument footprint than the snow pickets used in this investigation could discriminate the 3-D evolution of the tracer gas plume. Large (~ cubic meter volume), high resolution representations of permeability are not practical with current technology but in the future would enable one to discriminate between advection and changes in diffusion rate due to permeability changes. Though not available in our tests, we would recommend others employ a blend of CO with standard air rather than $N_2$, which would then be essentially neutrally buoyant.

## 6 Data availability

Data from these experiments may be obtained by corresponding with the first author.

**Acknowledgements**

We thank Dr. Noah Molotch and Dr. Michael Durand for organizing deployments at Storm Peak Lab, Colorado. We thank Lisa Dilley and USFS for arranging deployments in Deschutes National Forest, Oregon. Dr. Ziru Liu and Rebecca Hochreutener assisted with the field deployments. Finally, we thank Dr. Samuel Morin and an anonymous reviewer for their meticulous reviews that substantially improved this manuscript.

**Author contribution**

S. Drake, J. Selker and C. Higgins designed the experiments and S. Drake performed them. S. Drake wrote the original manuscript. J. Selker and C. Higgins edited it.

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

Table 1. Summary of cases used for this analysis. Wind speed is at 1-m nominal height.  Layer density is the average snow density for a distinct surface snow layer. Horizontal picket spacing and CO sensor depth are listed by picket, ordered left to right as shown in the figures.

| Date | Location | Surface Layer Density kg m$^{-3}$ | Picket Spacing cm | CO Sensor depth cm | Case Id | Release Id | Release Volume L | Mean wind speed m s$^{-1}$ | Sigma wind speed m s$^{-1}$ | Meteorological and Surface Snow Conditions. Air Temperature; Crystal type; Size; Hardness |
|---|---|---|---|---|---|---|---|---|---|---|
| 03 Apr 2014 | Dutchman Flat, Oregon | 249 | 29, 36, 28 | 13, 14, 14, 14 | 1 2 3 | R1 R2 R3 | 2.00 1.33 2.67 | 1.64 2.45 1.86 | 0.92 1.51 1.10 | Winds, predominantly from the SW with arrival of a surface front. Intermittent snowfall through the night. $T_{AIR}$ ranged -1 to 1 ℃; faceted crystals; coarse; 2F |
| 04 Apr 2014 | Dutchman Flat, Oregon | 249 | 11.5, 13.5, 16 | 6.8, 7.0, 6.0, 6.8 | 4 5 6 | R3 R4 R5 | 9.50 6.00 9.50 | 2.55 2.62 2.91 | 1.22 1.22 1.52 | Winds turning from SW to NW through the day. Low directional variability. Clearing weather through day. $T_{AIR}$ ranged -1 to 0 ℃; faceted crystals; coarse; 2F |
| 19 Apr 2014 | Santiam Pass, Oregon | 430 | 15.0, 17.6, 17.0 | 9.0, 9.6, 9.0, 9.0 | 7 8 9 10 11 | R1 R2 R3 R4 R5 | 2.30 4.00 5.00 6.67 1.17 | 2.83 2.19 1.68 0.75 0.38 | 1.47 1.10 0.92 0.56 0.22 | Clear day.  $T_{AIR}$ ranged -2 to 5 ℃; rounded grains, very coarse; 4F |
| 24 Mar 2015 | Storm Peak Lab, Colorado | Variable, ice layer | 15.5, 14.5, 15.0 | 21.0, 19.0, 16.0, 17.0 | 12 | R1 | 0.20 | 4.05 | 1.40 | Stormy. $T_{AIR}$ ranged -7 to -5 ℃ faceted crystals; medium; 3F |
| 26 Mar 2015 | Storm Peak Lab, Colorado | 227 | 19.0, 19.0, 18.0 | 18.0, 19.0, 19.0, 19.0 | 13 14 | R1 R2 | 0.20 0.20 | 3.41 3.06 | 0.92 1.07 | Clear day.  $T_{AIR}$ ranged -5 to -2 ℃; faceted crystals; medium; 2F |

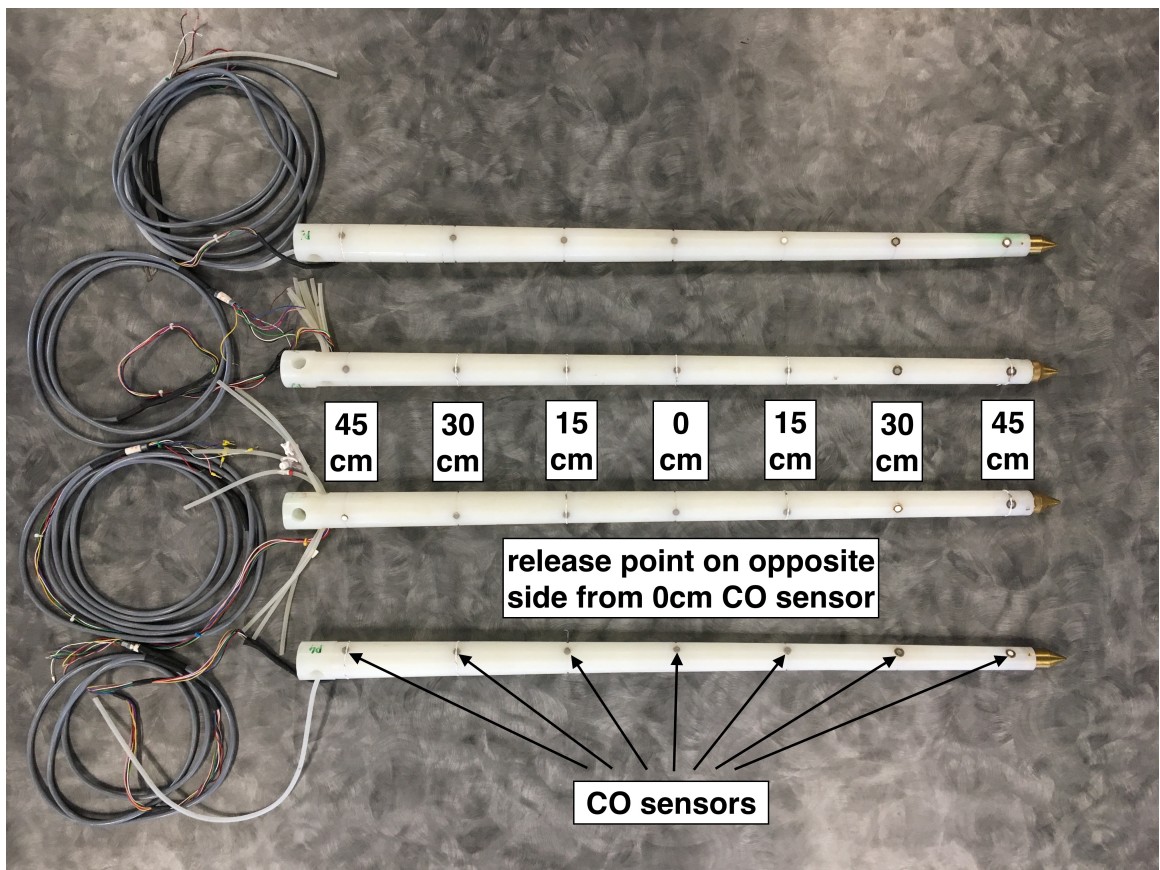

**Figure 1. Four 1-m snow pickets, each mounted with 7 CO sensors at 15-cm spacing. Along picket distances of the CO sensors from the central CO sensor are labelled in the picture. Carbon monoxide was released through tubing that exited the picket opposite the central (0cm) CO sensor on one of the pickets.**

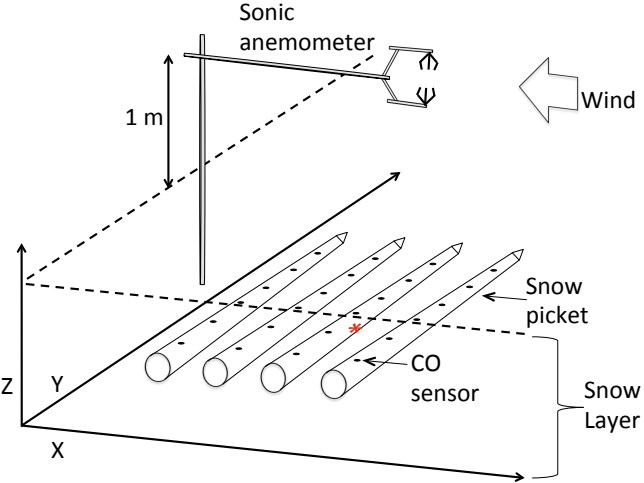

**Figure 2. Diagram of the experimental setup showing relative positions of the sonic anemometer and snow pickets. The red asterisk marks the release point on the bottom of one of the snow pickets. The CO sensors on four snow pickets define an x-y plane that is retained in subsequent figures.**

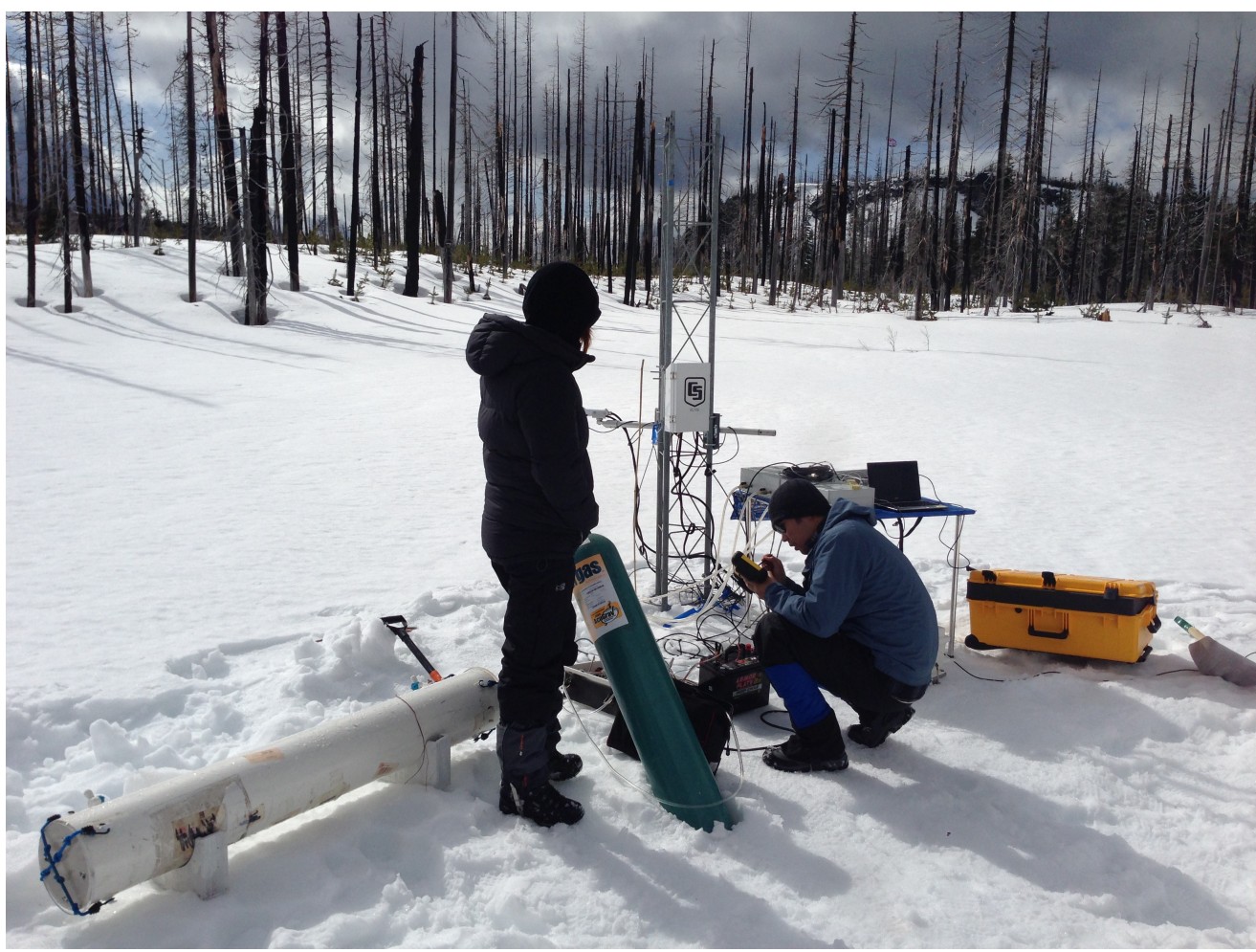

**Figure 3. Picture of the calibration chamber (white PVC conduit) at the Santiam Pass site. Snow pickets are placed in low-profile collars to keep them separated within the chamber. Silicone is used to seal the chamber endplates and duct seal is used to seal air gaps around the CO sensor wiring as it passes through the endplates. A dilute mixture of CO gas (green tank) is introduced into the chamber through an Aalborg GFM17 Mass Flowmeter and mixed by a fan within the chamber. CO concentration as measured by each sensor is acquired on a CR1000 logger (mounted in a weather resistant case) for ~ 10 minutes at each concentration measured. After each concentration measurement, CO gas mixture is evacuated to minimize sensor saturation.**

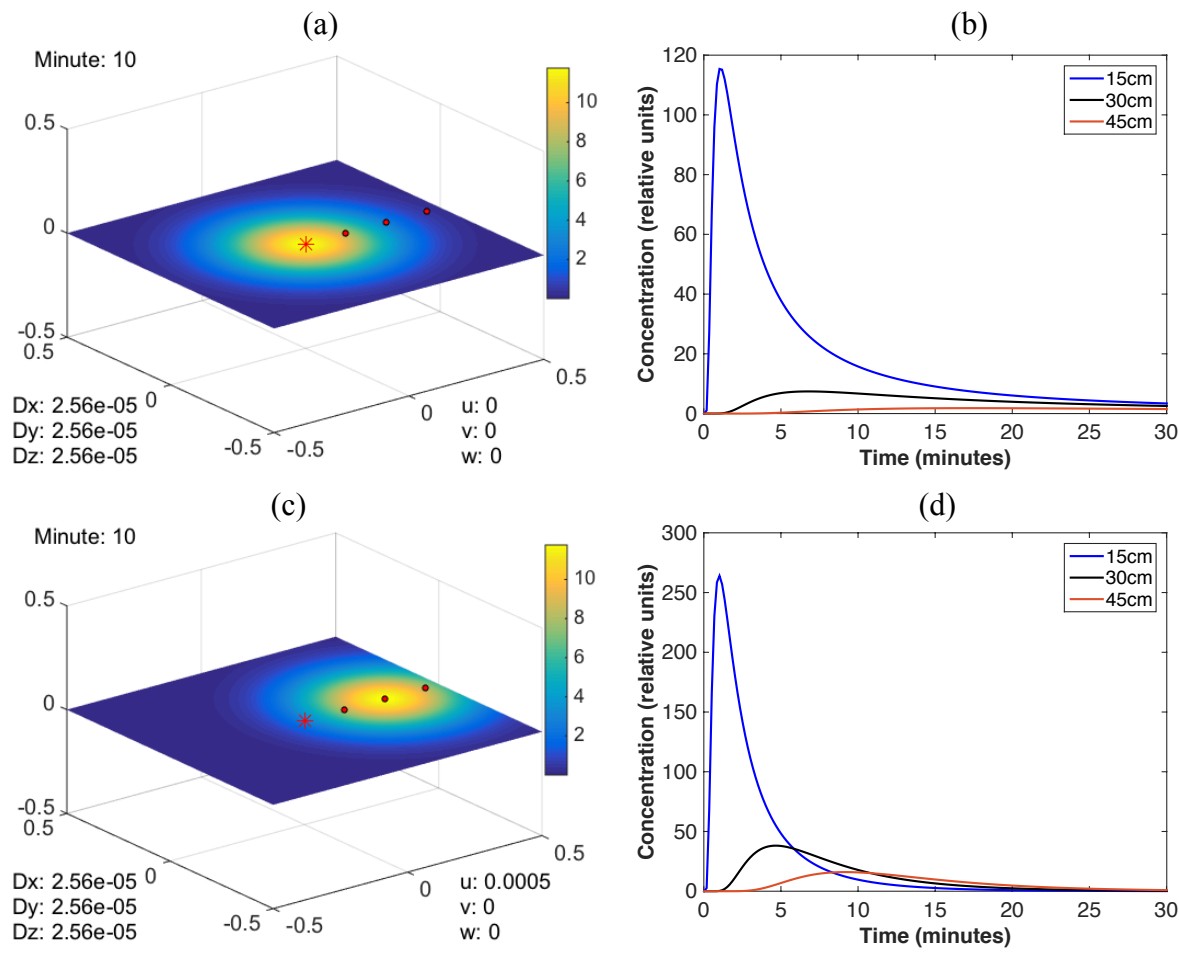

**Figure 4. Vertical cross-section of simulated plume dispersion in a purely diffusive case over a 1 m³ volume (panel *a*, upper left) and the associated breakthrough curve (panel *b*, upper right) and for a diffusive/advective case (panel *c*, lower left) with associated breakthrough curve (panel *d*, lower right). The red asterisk in panels *a* and *c* mark the plume release point. For the advective/diffusive scenario, plume concentration is greater at the 30-cm position than the 15-cm position after 10 minutes.**

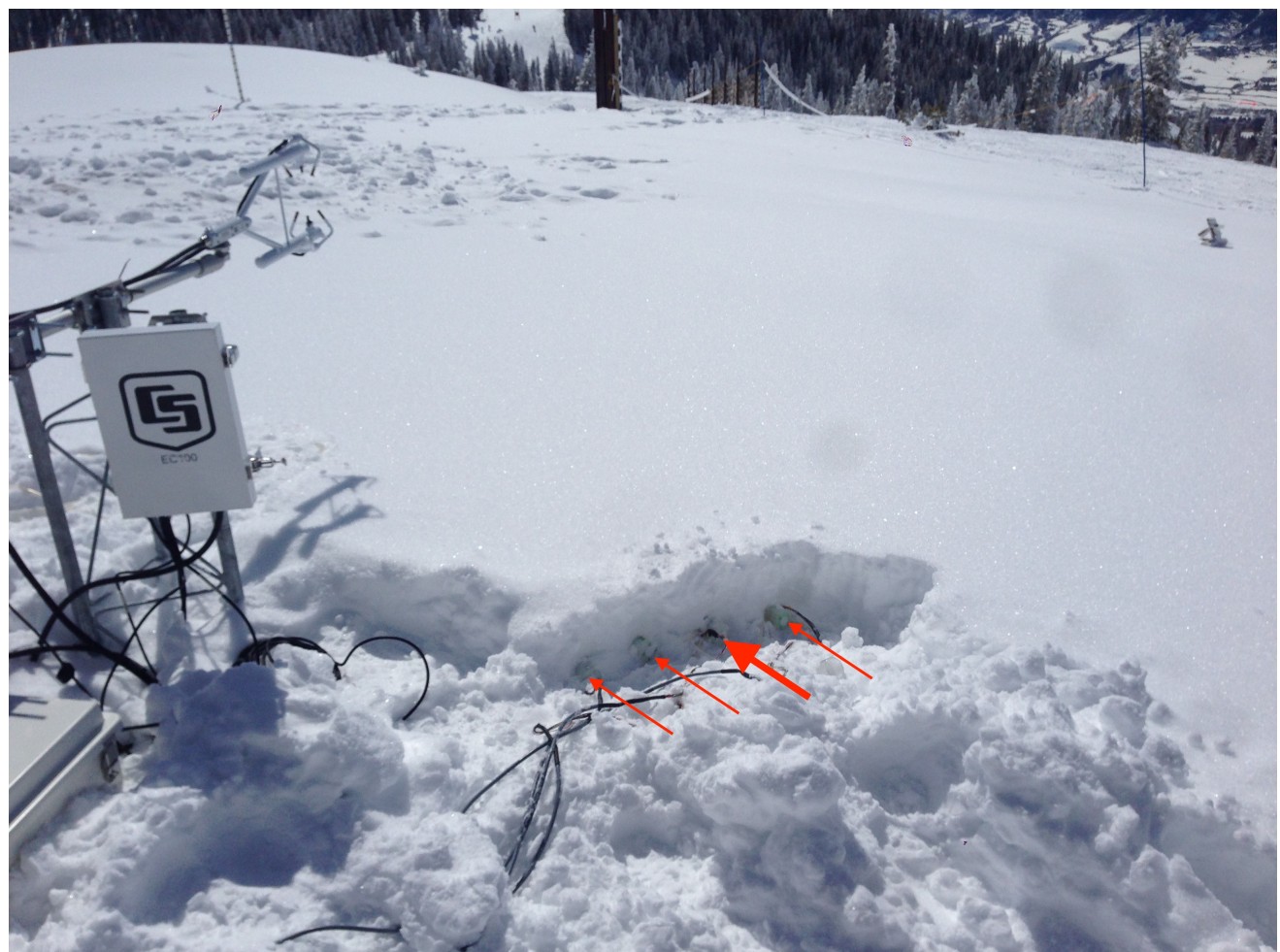

**Figure 5. This picture shows the experimental setup for cases 13 and 14 and was taken after the data acquisition period. Red arrows point to the tops of the snow pickets after we removed snow to expose the top and left side of the instrumented snow layer for documentation purposes. The bold red arrow points to the snow picket where the trace gas was released for these two cases. A Campbell Scientific Irgason mounted above the snow pickets acquired horizontal and vertical wind speed and direction at 20 Hz.**

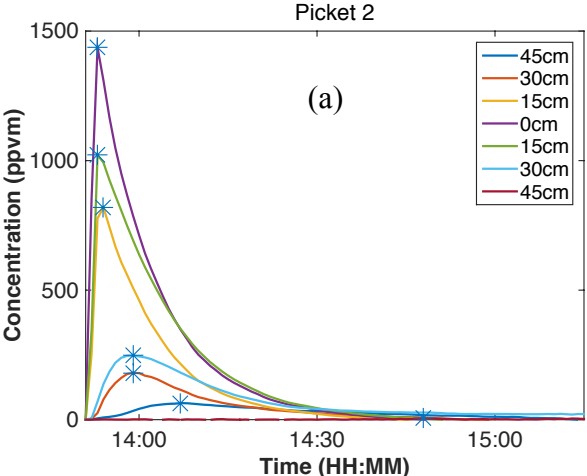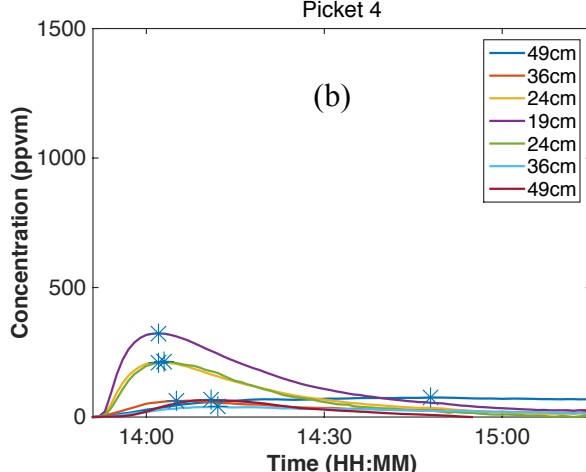

**Figure 6.** Calibrated near-field break-through curves for case 14 (panel *a*) and 19 cm downstream of the release point (panel *b*). CO concentration data were acquired at 1-minute resolution. The CO gas release position is on the opposite side of the picket from the mid-picket (0 cm) sensor in panel *a* (see also Fig. 2). Distances given in the legend are measured from the release point to the given sensor for both figures. Blue asterisks demark time of maximum concentration at that position.

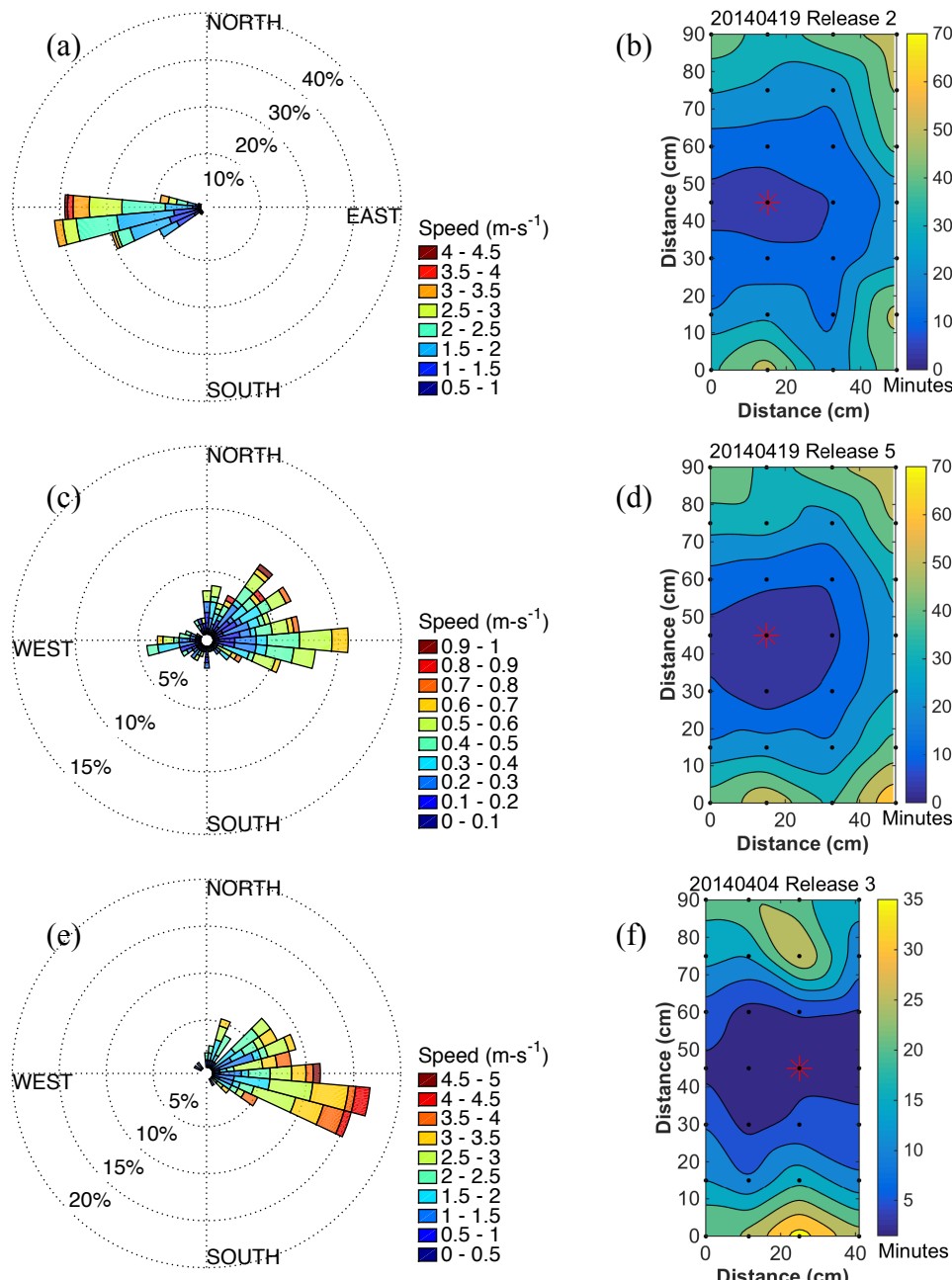

**Figure 7. Windrose for persistent wind case 8 in 430 kg-m$^{-3}$ snow (a), for light, variable wind case 11 also in 430 kg-m$^{-3}$ density snow (c) and for persistent wind with 249 kg-m$^{-3}$ density snow case 4 (e). Winds are plotted relative to pickets such that the tops of plots (b), (d) and (f) align with the north direction of the windroses. Wind speed is color-coded with speed ranges given in the legend. Time to maximum concentration is color-filled by minutes from release time for case 8 (b) and case 11 (d) and case 4 (f). Contours are in 10-minute increments for (b) and (d) and 5-minute increments for (f). CO sensor positions are marked as black dots and release point with red asterisks.**

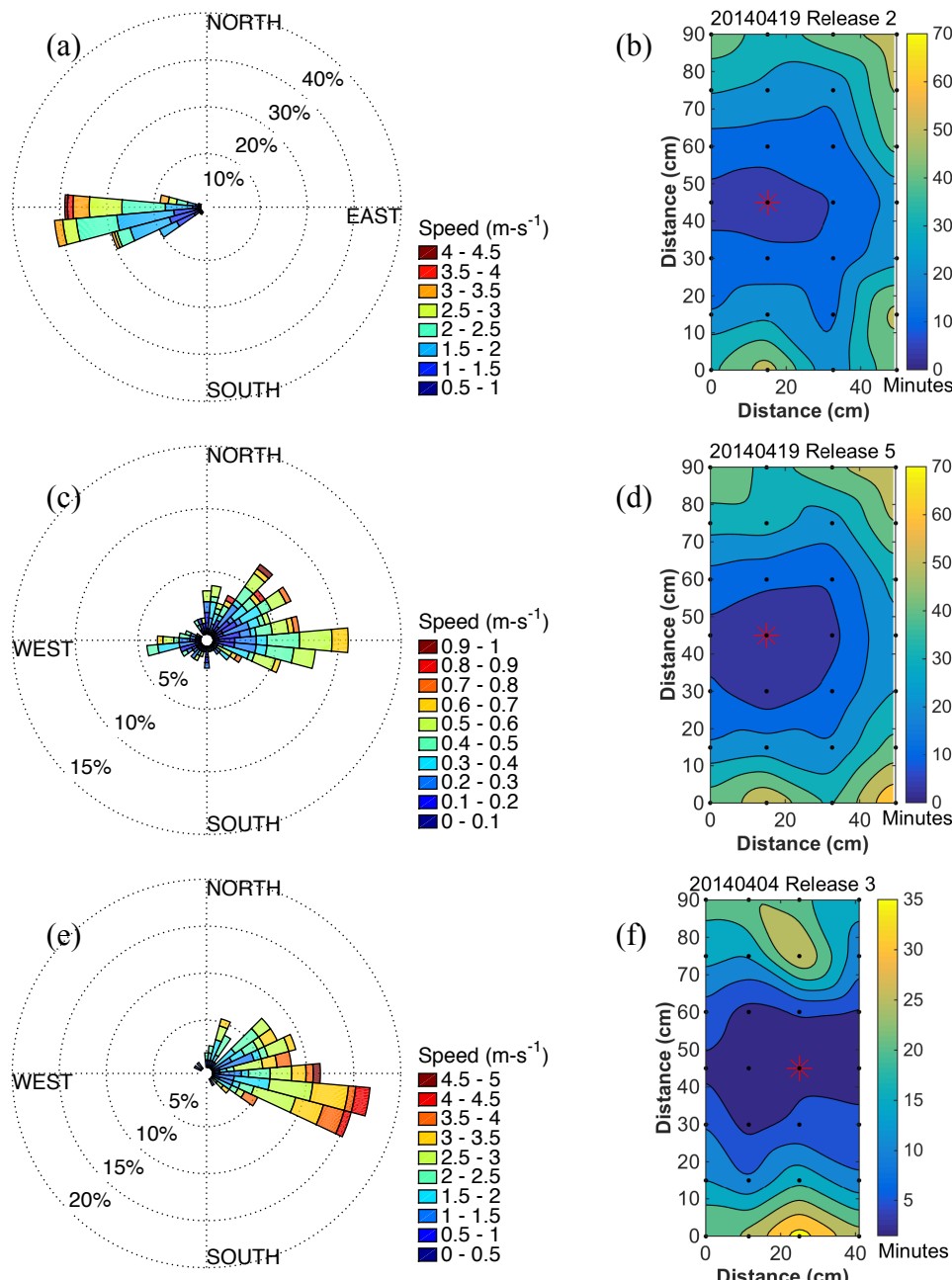

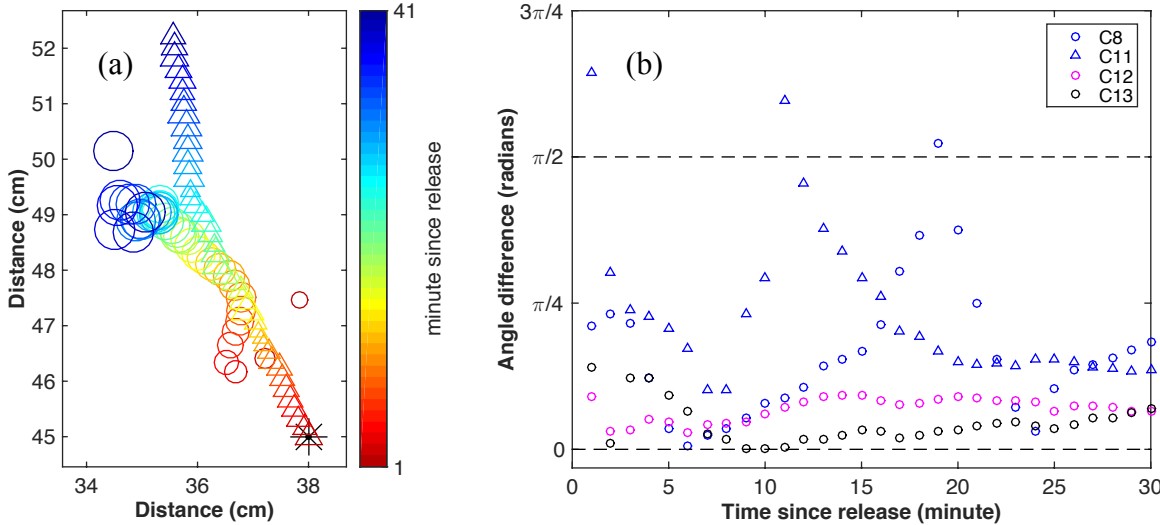

**Figure 8. Centroid of mass plotted for case 13 color-coded by minute since release (a). A black asterisk marks release position. In (b) the angle between the centroid of mass translation direction and the wind direction has higher correspondence in lower density snow (e.g., case 13).**

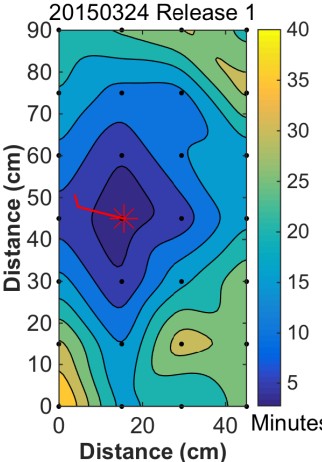

**Figure 9. Plot of the time required for CO to reach maximum concentration for a release with snow pickets placed below an ice layer. The red wind barb indicates the average wind speed (in knots) and direction during the measurement period (see also Table 1). The distance is relative to the bottom and leftmost sensor relative to a coordinate plane viewed from above. The plume has a molecular diffusion signature (roundish), a wind direction signature (downwind propagation) but also is characterized by flow along preferred pathways, which render as irregular lobes at the sensor network resolution.**

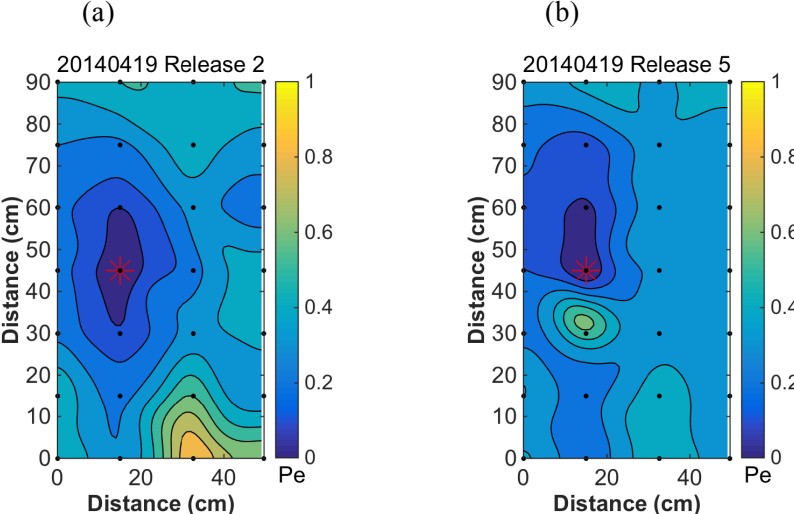

**Figure 10. Normalized Péclet number for windy case 8 (panel a) and low-wind case 11 (panel b). Both cases were obtained during the same picket deployment. Case 8 shows preferential streamwise dispersion (see also windroses in Fig. 7) as well as enhanced dispersion along high permeability pathways. In (b), but for the anomalously large Péclet number gradient just below the release point (due to a large release volume), the gradient is weak relative to (a), characteristic of a diffusion-dominated regime.**

