# Peer review of "Wind enhances differential air advection in surface snow at submeter scales"

_The Cryosphere, 2017_

## Referee Comment (RC1) · Anonymous Referee #1 · 24 Apr 2017

eneral comments

This paper demonstrates an interesting method to better understand the relevance of advective flow in a snowpack. This is demonstrated by several measurements. What was a bit surprising to the reviewer that the method is already described in much detail by Huwald et al., 2012. The paper demonstrates the application of the method in several field cases. The authors demonstrate that diffusion and advection of carbon monoxide is affected by wind speed. The main conclusion of the paper is that "atmospheric pressure gradients can induce subsurface advection". This is not an entirely new results. Unfortunately, the physical properties of the investigated snowpacks are not described to a degree that is state-of-art. Neither a highly resolved density profile, specific surface area measurements, measurements of spatial variability (using e.g. near-infrared photography, Tape et al. 2010) were applied. The use and progress of

this paper for interpretation of advective flows in snow is therefore very limited beyond description. The presented model, assuming isotropic and non-layered properties of the snowpack, is very simplistic.

Specific comments Title: Snow is always porous and air is a constituent, and there is no closed porosity (the major difference to firn). So "interstitial air" is redundant. A better fitting title could be: "A trace gas method of evaluating macroscopic air advection and diffusion in snow"

page 2,line 4: More recent measurements show that basic properties of the snowpack do change often in a very complex way within one layer. The traditional method to characterize a snowpack requires usually cast samples (e.g. Arakawa et al.) or other recently developed quantitative techniques.

page 2, line 15: The backfilled snowpit (dimensions?) could be a major source of disturbance for the measurements, as the density (and consequently permeability) is easily increased by about 20%. Any checks or numerical simulations of this effect?

page 2, line 20: What is "relatively high-density, spongy snow"? Which method was used (beyond interpretation of the measurements) to assure that no leakage occurred?

page 4, line 1: how was homogeneity measured? A single storm event can easily create several mm-thick denser layers.

page 5, line 15 ff: Riche and Schneebeli (2012) measured enhanced horizontal thermal conductivity in snow with little or no temperature gradient metamorphism. This would contradict the general statements about the snowpack in his paper. Clearly, anisotropy at several scales (mm to dm) is a key factor for diffusive processes.

page 7, line 28 (and other places) please define "low density snow" "high density snow" in quantitative terms.

page 9, line 19: The conclusion drawn here is not well supported. The observed pattern (especially Fig. 8)= is in my view not at all conclusive (the point x=20,y=30 could be an

outlier).

Table 1: Definitions of the snowpack are insufficient for any comparison or application of the results. How was the density measured? What was the vertical spacing? What was the snow type (see International Classification) etc.

Table 2: What was the snow temperature / temperature profile? The description of the snow seems to indicate that the snow stratigraphy was rather complex (guess ...)

Searching for papers about this topic, I found the following references which seem to be relevant to the topic:

Massman, W. J., and J. M. Frank (2006), Advective transport of $CO_2$ in permeable media induced by atmospheric pressure fluctuations: 2. Observational evidence under snowpacks, J. Geophys. Res., 111(G3), 1–11, doi:10.1029/2006JG000164.

Ebner, P. P., M. Schneebeli, and A. Steinfeld (2015), Tomography-based monitoring of isothermal snow metamorphism under advective conditions, Cryosph., 9(4), 1363–1371, doi:10.5194/tc-9-1363-2015.

Massman, W. J. (2006), Advective transport of $CO_2$ in permeable media induced by atmospheric pressure fluctuations: 1. An analytical model, J. Geophys. Res., 111(G3), 1–14, doi:10.1029/2006JG000163.

Ebner, P. P., M. Schneebeli, and A. Steinfeld (2016), Metamorphism during temperature gradient with undersaturated advective airflow in a snow sample, Cryosphere, 10(2), 791–797, doi:10.5194/tc-10-791-2016.

Ebner, P. P., C. Andreoli, M. Schneebeli, and A. Steinfeld (2015), Tomography-based characterization of ice-air interface dynamics of temperature gradient snow metamorphism under advective conditions, J. Geophys. Res. Earth Surf., 120(12), 2437–2451, doi:10.1002/2015JF003648.

Other references: Tape, K. D., N. Rutter, H.-P. Marshall, R. Essery, and M. Sturm

(2010), Recording microscale variations in snowpack layering using near-infrared photography, J. Glaciol., 56(195), 75–80, doi:10.3189/002214310791190938.

Arakawa, H., K. Izumi, K. Kawashima, and T. Kawamura (2009), Study on quantitative classification of seasonal snow using specific surface area and intrinsic permeability, Cold Reg. Sci. Technol., 59(2), 163–168, doi:10.1016/j.coldregions.2009.07.004.

Calonne, N., M. Montagnat, M. Matzl, and M. Schneebeli (2017), The layered evolution of fabric and microstructure of snow at Point Barnola, Central East Antarctica, Earth Planet. Sci. Lett., 460, 293–301, doi:10.1016/j.epsl.2016.11.041.

---

## Referee Comment (RC2) · S. Morin (Referee) · 1 May 2017

The discussion article entitled "A trace gas method of evaluating interstitial air advection and diffusion in snow" provides potentially useful data and analysis in order to address the partitioning between the various processes responsible for trace gas movement in snow. This topic is fully consistent with the scope of The Cryosphere, and requires careful experimental investigations combined with modelling approaches, which are both employed in this study. Unfortunately, the manuscript lacks precision in many respects, in particular including the description of the field experiments and numerical modelling, the terminology used, the physical properties of snow, and the literature references. In general more quantification is needed in the text, which is often too vague and qualitative (although actual numbers can sometimes be found in the tables and figures, but unfortunately not used in the text).

[Figure]

The only general comment I have refers to the description of the processes conducive to tarce gas movements in snow. The title refers only to "advection" and "diffusion". It seems, from reading parts of the article, that the authors make a distinction between different advection processes (at least two : "quasi-static pressure gradients [...] in response to wind-induced pressure gradients"– page 1, line 19, "turbulently generated pressure fluctuations" – page 1, line 22). Given that the literature in this field is complex and there may be significant overlap between concepts used in the literature, it would be useful that this manuscript describes the processes very accurately and precisely, in order to avoid ambiguity in this central issue for this manuscript.

Besides this need for clarification, I do not have any major comment on the manuscript, having not detected any major flaw, but conversely it is often very challenging to precisely understand how the experiments (numerical and in the field) were designed, how the results were achieved, and what are their implications. I thus rather than major remarks have quite a long list of comments and points, which deserve attention from the authors in order to better convey their research to the readers. Not being myself a native english speaker, I refrain from any comments on the wording, although in several places the wording seems (perhaps) colloquial and ambiguous (I hope this can be addressed post-acceptance through the copy-editing step).

Nevertheless, I believe that a significantly improved manuscript could be accepted for publication, given the relevance of the experiments carried out and their potential to be converted into useful scientific results.

Specific comments

Title

I think it would be appropriate to mention in the title that the emphasis is placed here on "wind-driven" processes.

Abstract

Page 1, line 9 : I recommend rephrasing the term "validated". It is very unclear what is meant here. "Validate" is a very strong verb.

Page 1, line 14 : Besides "surface (chemical) reaction rates and interpretation of firn and ice cores", I think the primary implications of this work apply to interstitial water vapor movement and snow metamorphism itself, see e.g. Calonne et al. (2015). This is unfortunately almost never mentioned in this manuscript.

Introduction

The introduction is short and does not provide a sufficiently broad status of the current knowledge in this field. For example, on the experimental work it ignores work carried out in order to measure fluxes of trace species in snow (e.g. Seok et al., 2009). On the snow microstructure side, it only vaguely alludes to some of the physical processes and properties involved (e.g. page 2, line 4 the word "permeability" is mentioned but the introduction lacks the establishment of the conceptual and physical framework where such variables are used). Without a proper introduction of the state of the literature and knowledge gaps, it is difficult to understand what added-value is brought by the current manuscript. I strongly recomment expanding the part from page 2, line 5 to 9 in order to better describe what previous knowledge gap is addressed by the current study, and how. At present, the introductory statement regarding the current study seems only to be a disclaimer on why only top-most snow layers are targeted, without even knowing what kind of measurements are dealt with and to serve what purpose.

Methods

In general the "Methods" section needs considerable improvements, to enhance the clarity of the description of the experimental goals, the instruments used for CO measurements but also ancillary conditions (not only wind, but also snow properties, meteorological conditions etc.) and the experimental sites. Figure 2 is very unclear, I recommend providing pictures of the set-up in the field and conceptual sketches illustrating the actual set-up for the various configurations used (apparently implementations vary
from the various cases reported). Maybe such sketches should be provided along with every result graphically shown, in order to better understand the set-up and the corresponding data. I think the first sentence of the "Methods" belongs to the Introduction (in order to better explain the goals of the manuscript), and everything else should be placed in sub-sections with informative title. The status of the part from page 2, line 10 to page 3, line 5, is unclear and often repeated with the content of sections 2.1 and 2.2 (e.g. height of the wind sensor), which is illustrative of this confusion.

Specifically:

- neutral buyoancy : The paragraph from page 2, line 10 to line 22 is quite tortuous and mixes several issues together. Line 15 it is said that "it is nearly neutrally buyoant", then line 18 "neutral buoyancy is not strictly achieved for this experiment" then line 22 recommendations are made to make the experiment "essentially neutrally buyoant". It is very hard from this text what is really at stake here, given that no quantification is provided and the text is meandering around the issue of neutral buyancy.

- safety : The same paragraph states line 16 that CO is "safely handled" then line 20 it can "cause unhealthy side effects". Here again, better clarity is required.

- page 2, line 17 : reference needed here.

- page 2, line 20 – 22 : I think such "recommendations" could be placed in the Discussion or Conclusions sections (along with a quantification of the related issues), not in the first paragraph of the Methods section.

- page 2, line 32 : "cases 12 through 14": at this point, the "cases" have not been introduced yet. This illustrates the need for a major reorganisation of the manuscript, in order to better streamline the description of the methods and experiments.

- Sections 2.1 and 2.2 do not seem logically organized. For example, in section 2.1, page 3, lines 13 to 20 describes the experimental approach employed to deploy the sensors and contains details relevant to the equilibration time, which could be referred
to as "Deployement description", and not at all a description of the sites. Conversely, section 2.2 confuses data selection (first words of the section), quality control and operational constraints, which do not correspond a a description of the deployment. This calls for a major reorganization of the Methods section.

- section 2.1, page 3, line 9 : more information should be given on what is referred to the "broad range of wind forcing and snow permeability regimes".

- section 2.1, page 3, line 13: what is a "low-profile snowpit" ?

- section 2.1, page 3, line 20: what is "spongy snow" ? If this corresponds to a description of snow properties, this should be provided in the Results section, while the Methods section should describe the methods employed to characterize snow physical properties (currently missing, although this is critical for this study).

Section 2.3

Page 4, line 5: that CO sensors are sensitive to humidity and temperature is already mentioned above.

Page 4, lines 9 and 10 : perhaps a sketch and pictures could be useful to better understand the functioning of the calibration chamber. At present, this is very unclear and there is a large margin for interpretation of the sentences describing the apparatus.

Page 4, line 15 and 16 : what is the definition for "cold snow" and "warm snow" calibration ? How is this implemented in practice for field measurements ?

Section 3 Data analysis

Page 4, lines 21 to 25: these statements are not consistent with the title of the section.

Page 4, line 30 : "modeled" needs to be replaced with "simulated" is what is dealt with here is actual simulation results. In addition, given that the equation used (the model) includes advection and diffusion (line 31), how is it possible that deviations between observations and simulations correspond to the influence of "advection and

snow heterogeneity" (line 30) ? This is very unclear and needs to be better described.

Page 4, equations (1), (2) and (3) : symbols need to be described in the text. What is D, r, t, C etc. ? What is tMAX ? Thye absence of description of the symbols hampers the understanding of the reste of the manuscript, unfortunately.

Page 4, line 10 : I con't understand whay is meant by "We calculated the diffusion coefficient for each sensor". The sensors measure the concentration of CO, I don't see how this solely can be used to compute the diffusion coefficient.

Section 3.1 (note that, if there is only one subsection at the end of section 3, this implies that the structure is not optimal; either add more subsections from the beginning of the section 3, or drop the section 3.1 and make simple paragraph).

Page 5, line 15 ; it is surprising that a reference to Riche and Schneebeli (2012) is given to support the fact that snow permeability could be anisotropic. First of all, anistropy of effective thermal conductivity was demonstrated by Calonne et al. (2011), but, more importantly, Calonne et al. (2012) provide direct estimates of the anisotropy of the intrinsic permeability of snow for various snow types fould in mid-latitude snow types. There is thus no need to speculate here. Given the existence of literature not accounted for in this paragraph, it probably needs full rephrasing.

Results

Page 5, line 23 : what is a "diffusion constant" ?

Page 6, line 10 : "moderate" : rather provide numbers. "mid-to-low density" : rather provide numbers.

Page 7, line 6 : what is the "plume standard deviation" ? Line 7 : what do "minutely" refer to ? Earlier in the manuscript this refers to "1-minute time resolution".

Page 7, line 14: "NaN" should be defined.

Page 7, line 19 : is there an analytical form for Equation (1), which accounts for nonhomogeneous diffusion coefficients ?

Page 7, line 23 : "higher permeability (lower density)" : better provide numbers. Furthermore, as demonstrated in Calonne et al. (2012), not only density but also specific surface area, drive variations of snow permeability.

Page 7, line 30 : "conspired" ?

Page 8, line 23 : How was the diffusivity "measured" ? Up to this point in the manuscript, no apparatus measuring the diffusivity was described.

Page 9, line 2 : what is the "smallest measured effective diffusivity" ? Over all measurements across sites ? What is the value found ? How does it compare with literature values ? This is not clear.

Page 9, line 3 : note that equation (3) already exists in the manuscript (this should be equation (4)).

Conclusions

Page 9, line 21 : "invalidating the notion of a mono-valued diffusion coefficient over small areas" : this notion was never introduced before in the article. This statement requires that the literature review identifies the need to "invalidate" (or not) this "notion".

Page 9, line 29 : "gran-scale properties" : this is the first mention of microstructure-scale snow properties. This aspect deserves to be better introduced in the manuscript, on the basis of references suggested here as well as in the review provided by Reviewer #1. Otherwise, this manuscript will be inconsistent with current knowledge in snow physics.

Authors contribution:

Page 10, line 10 : Given that Z. Liu and R. Hochreutner are not authors of this manuscript, I don't understand why they are mentioned here (it is of course appropriate to mention them in the Acknowledgements).

Tables and Figures:

Table 1 : More information on snow stratigraphy is needed, given that this concerns only 5 snow pits this should be doable in a condensed form. Also, in addition to mean wind-speed, its standard deviation would be useful to better assess the steadiness of the wind conditions. The content of the CO sensor depth column is not clear (and it has formatting issues, some number have a '.0", some not).

Table 2 : Could be merged with Table 1. "Degree" symbol missing.

Figure 1 : Size scale missing.

Figure 2 : Very unclear. I don't understand where is the CO cylinder and where the CO is injected into snow. The different arrangements of the "cases" could be better explained, maybe using a "3D" sketch (yet simple) which would better show how the apparatus was implemented in the field. Pictures could be used to better illustrate how the system was implemented.

Figure 3 : The CO release position should be indicated on panels a and c

Figure 4 : I could not understand why there are several lines with the same "distance" (e.g. twice 45 cm in subplot a). Only a sketch explaing the set-up of this particular experiment could help, I'm afraid. As such it is highly ambiguous.

Figure 5 : "Distances" in the labels of the axes in b, d and f should be better described (are these horizontal or vertical distances ? Where is the injection of CO (is is the red star ?) ? Again, a sketch describing this experiment would be more than useful.

Figure 6 : "Distances" in the labels of the axes in subplot a should be better described (see comments above).

Figure 7 : Same comments as for Figure 5 . Furthermore, what is the red line ?

Figure 8 : Same comments as for Figure 7. The caption refers to "impedence" which appears unique in the manuscript and is not referred to in the text.

References (if not already quoted in the manuscript)

Calonne, N., Flin, F., Morin, S., Lesaffre, B., du Roscoat, S. R., and Geindreau, C.: Numerical and experimental investigations of the effective thermal conductivity of snow, Geophys. Res. Lett., 38, L23501, doi: 10.1029/2011GL049234, 2011.

Calonne, N., C. Geindreau, F. Flin, S. Morin, B. Lesaffre, S. Rolland du Roscoat and P. Charrier, 2012. 3-D image-based numerical computations of snow permeabilityÂă: links to specific surface area, density, and microstructural anisotropy, The Cryosphere, 6, 939-951, doi: 10.5194/tc-6-939-2012, 2012.

Calonne, N., C. Geindreau, F. Flin, 2015. Macroscopic modeling of heat and water vapor transfer with phase change in dry snow based on an upscaling methodÂă: Influence of air convection, J. Geophys. Res.Âă: Earth Surf., 120, 2476-2497, doi: 10.1002/2015JF003605, 2015.

Seok, Brian, Detlev Helmig,ÂăMark Williams,ÂăDaniel Liptzin,ÂăChowanski, K.,ÂăJacques Hueber: An automated system for continuous measurements of trace gas fluxes through snow: An evaluation of the gas diffusion method at a subalpine forest site, Niwot Ridge, Colorado. Biogeochemistry, 95(1): 95-113. doi: 10.1007/s10533-009-9302-3 , 2009

---

## Author Comment (AC1) · 7 Jun 2017

Tuesday, June 6, 2017

Author's response to Reviewer #1 comments

The lead author will address your comments. Author responses reference the same page and line number as written in the original manuscript. Due to the detailed and sometimes overlapping changes requested by both reviewers we summarize all document changes as a PDF supplement.

General comments This paper demonstrates an interesting method to better understand the relevance of advective flow in a snowpack. This is demonstrated by several measurements. What was a bit surprising to the reviewer that the method is already

described in much detail by Huwald et al., 2012. The paper demonstrates the application of the method in several field cases. The authors demonstrate that diffusion and advection of carbon monoxide is affected by wind speed. The main conclusion of the paper is that "atmospheric pressure gradients can induce subsurface advection". This is not an entirely new results. Unfortunately, the physical properties of the investigated snowpacks are not described to a degree that is state-of-art. Neither a highly resolved density profile, specific surface area measurements, measurements of spatial variability (using e.g. near-infrared photography, Tape et al. 2010) were applied. The use and progress of this paper for interpretation of advective flows in snow is therefore very limited beyond description. The presented model, assuming isotropic and non-layered properties of the snowpack, is very simplistic.

Author's response to general comments:

Thank-you for your thoughtful review of this manuscript. I'll address your general comments one at a time.

"What was a bit surprising to the reviewer that the method is already described in much detail by Huwald et al., 2012."

Author response: Excluding instrumentation repair that involved rewiring and replacing 6 CO sensors, the CO measurement instrumentation described in this manuscript is the same as described by Huwald et al., 2012. The deployment design of the snow pickets is different, however. In Huwald et al., 2012, snow pickets were placed vertically with the picket ends protruding from the snow surface, introducing experimental problems with which the authors of that paper needed to contend. For example, wind blowing over the tops of the pickets created a pressure gradient that exacerbated along-picket leakage. As stated in the methods section, we alternatively dug a shallow trench to expose a face of the surface snow layer and pushed the snow pickets horizontally and parallel to each other into this undisturbed surface snow layer. After placement, we backfilled the shallow trench to completely cover the exposed ends of the snow

pickets. The revised version of the manuscript will delineate these points more clearly.

"The paper demonstrates the application of the method in several field cases."

Author response: The number of deployments was purposefully limited to discrete snowfall events that were several times deeper than the snow picket diameter so that the snow picket would not unduly influence CO plume evolution. This experimental design choice limited the number of possible deployments.

"The main conclusion of the paper is that 'atmospheric pressure gradients can induce subsurface advection'. This is not an entirely new result."

Author response: The main conclusion of the manuscript is that the evolution of a trace gas plume in snow under windy conditions reveals an advective signature as well as the presence of preferential pathways. Previous work has inferred gaseous advection using temperature measurements (Albert and Hardy, 1995, Sokratov and Sato, 2000) and point-based measurements of tracer gas concentrations (Albert and Shultz, 2002). What is innovative here is that we have explicitly resolved advection in a plane (e.g. 2D) using tracer gas.

"Neither a highly resolved density profile, specific surface area measurements, measurements of spatial variability (using e.g. near-infrared photography, Tape et al. 2010) were applied."

Author response: Indeed, the suggested field-based measurements of snow characteristics would improve context for the results. However, there is a scale mismatch between the high-resolution sampling suggested by the reviewer and the representative volume of the tracer gas measurements. For example, a highly resolved density profile of the 1m $\times$ 1m $\times$10cm domain using a 100 cc Hydro-tech sampler would have required 1000 measurements. Optimistically assuming that no samples were damaged, at 1 minute per sample this would have required 16.7 hours to complete and still would not describe the snow with fine enough resolution to model airflow for a particular snow
state. Near-IR photography gives fine-scale resolution of a 2D section but not in 3D and not with the precision needed to represent snow topology accurately enough to model differences in airflow in the presence of weak and/or transient pressure gradients. The authors are not aware of a successful cast of a snow sample this large (approximately 1m × 1m × 20cm) by current techniques. The snow samples in Calonne et al. (2012) were cubes ranging from 2.51mm to 9.16mm per side. More than 105 samples of the 9.16mm per side cube would be needed to describe the deployment volume. In short, a method to image a volume of snow on the order of that used in this experiment with sufficient precision to simulate fine-scale airflow through natural snow has not yet been demonstrated.

"The use and progress of this paper for interpretation of advective flows in snow is therefore very limited beyond description."

Author response:

Quantitative analysis is limited because we characterize air movement in two dimensions rather than three. A lack of concentration measurement in the vertical coordinate limited the quantitative results because we could not resolve vertical mass flux. Nevertheless, we do quantify how the normalized Peclet number changes between a windy and calm case. Technological improvements in imaging larger sections of snow structure in 3D would facilitate quantitative analysis and model simulations.

"The presented model, assuming isotropic and non-layered properties of the snowpack, is very simplistic."

Author response:

The simplicity of an analytical advection-diffusion model is precisely what makes it an effective standard that, when compared with real-world experimental results, highlights the deviation of an experimental result from an idealized scenario. This technique allows us to discriminate deviations from isotropy that characterize real-world snowpack

state.

Specific comments

Title: Snow is always porous and air is a constituent, and there is no closed porosity (the major difference to firn). So "interstitial air" is redundant. A better fitting title could be: "A trace gas method of evaluating macroscopic air advection and diffusion in snow"

Author's response to title:

Thank you for this suggested change. Our intention was to emphasize that the manuscript addresses air contained within the top layer of the snowpack. We agree with your assertion and changed the title.

page 2,line 4: More recent measurements show that basic properties of the snowpack do change often in a very complex way within one layer. The traditional method to characterize a snowpack requires usually cast samples (e.g. Arakawa et al.) or other recently developed quantitative techniques.

Author's response to page 2,line 4:

(This issue was addressed above in the general comments section.)

page 3, line 15: The backfilled snowpit (dimensions?) could be a major source of disturbance for the measurements, as the density (and consequently permeability) is easily increased by about 20%. Any checks or numerical simulations of this effect?

Author response to page 3, line 15:

The snow pit merely provided access by which to place the snow pickets horizontally into undisturbed snow. The CO measurements were acquired in undisturbed snow. The snow pit was backfilled to minimize surface roughness gradients that might generate a pressure field that could influence CO plume evolution. The snow pit wall was parallel to the prevailing wind direction so as to minimize the disturbance to in-snow advection. The snow pit depth was equal to the surface layer of snow and the width
was approximately the 1-m length of the snow pickets. The manuscript was modified to clarify these points. For clarity, we changed the verbage from "snow pit" to "snow trench" in the manuscript.

Author's response to page 2, line 15:

page 3, line 20: What is "relatively high-density, spongy snow"? Which method was used (beyond interpretation of the measurements) to assure that no leakage occurred?

Author's response to page 2, line 20:

The average snow density of the snow layer in which the snow pickets were deployed is provided in column 5 of Table 1 for each case. The snow hardness is provided in Table 2 (since condensed to Table 1). We agree that the term "spongy snow" is vague and non-standard. We will instead use the density and hardness measurements to characterize the type of snow that seats snugly against the snow picket. Snow having a hardness of 1-finger seemed to form the best seal against the snow picket. Leakage around a picket was inferred by anomalously high propagation rate of the CO gas as detected by sensors along a single picket.

page 4, line 1: how was homogeneity measured? A single storm event can easily create several mm-thick denser layers.

Author's response to page 4, line 1

We changed "homogenous" to "discrete". Plume evolution of the trace gas releases reveal snow inhomogeneities. We revised the manuscript to clarify that we do not expect a snow layer to be homogenous.

page 5, line 15 ff: Riche and Schneebeli (2012) measured enhanced horizontal thermal conductivity in snow with little or no temperature gradient metamorphism. This would contradict the general statements about the snowpack in his paper. Clearly, anisotropy at several scales (mm to dm) is a key factor for diffusive processes. We rephrased this section. We concur that anisotropy is a key factor for diffusive processes and

the difference in length scale between pore space dimension and the length of, for example, a fracture poses modeling challenges for permeable media.

Author's response to page 5, line 15:

page 7, line 28 (and other places) please define "low density snow" "high density snow" in quantitative terms.

Author's response to page 7, line 28:

We changed this sentence replacing "low density snow" and "moderate wind speed" with values given in Table 1. page 9, line 19: The conclusion drawn here is not well supported. The observed pattern (especially Fig. 8)= is in my view not at all conclusive (the point x=20,y=30 could be an outlier).

Author's response to page 9, line 19:

We agree that the observed pattern in Fig. 8 at point x=20, y=30 is an outlier and examine a probable cause for this anomaly on page 9, line 9. The conclusion is not based on the value at x=20, y=30 but rather on the larger normalized Peclet numbers throughout the domain in Fig 8a relative to Fig 8b.

Table 1: Definitions of the snowpack are insufficient for any comparison or application of the results. How was the density measured? What was the vertical spacing? What was the snow type (see International Classification) etc.

Author's response to Table 1:

We added a description of how density was measured in the methods section. The snow type is shown in Table 2, column 3.

Table 2: What was the snow temperature / temperature profile? The description of the snow seems to indicate that the snow stratigraphy was rather complex (guess ...)

Author's response to Table 2:

In the updated manuscript we clarify that measurements are acquired in a single, thick surface snow layer so a detailed stratigraphic description and temperature profile below the layer in question is not relevant.

Searching for papers about this topic, I found the following references, which seem to be relevant to the topic: Massman, W. J., and J. M. Frank (2006), Advective transport of $CO_2$ in permeable media induced by atmospheric pressure fluctuations: 2. Observational evidence under snowpacks, J. Geophys. Res., 111(G3), 1–11, doi:10.1029/2006JG000164. Ebner, P. P., M. Schneebeli, and A. Steinfeld (2015), Tomography-based monitoring of isothermal snow metamorphism under advective conditions, Cryosph., 9(4), 1363– 1371, doi:10.5194/tc-9-1363-2015. Massman, W. J. (2006), Advective transport of $CO_2$ in permeable media induced by atmospheric pressure fluctuations: 1. An analytical model, J. Geophys. Res., 111(G3), 1–14, doi:10.1029/2006JG000163. Ebner, P. P., M. Schneebeli, and A. Steinfeld (2016), Metamorphism during temperature gradient with undersaturated advective airflow in a snow sample, Cryosphere, 10(2), 791–797, doi:10.5194/tc-10-791-2016. Ebner, P. P., C. Andreoli, M. Schneebeli, and A. Steinfeld (2015), Tomography-based characterization of ice-air interface dynamics of temperature gradient snow metamorphism under advective conditions, J. Geophys. Res. Earth Surf., 120(12), 2437–2451, doi:10.1002/2015JF003648. Other references: Tape, K. D., N. Rutter, H.-P. Marshall, R. Essery, and M. Sturm C3 (2010), Recording microscale variations in snowpack layering using near-infrared pho- tography, J. Glaciol., 56(195), 75–80, doi:10.3189/002214310791190938. Arakawa, H., K. Izumi, K. Kawashima, and T. Kawamura (2009), Study on quantitative classification of seasonal snow using specific surface area and intrinsic permeability, Cold Reg. Sci. Technol., 59(2), 163–168, doi:10.1016/j.coldregions.2009.07.004. Calonne, N., M. Montagnat, M. Matzl, and M. Schneebeli (2017), The layered evolution of fabric and microstructure of snow at Point Barnola, Central East Antarctica, Earth Planet. Sci. Lett., 460, 293–301, doi:10.1016/j.epsl.2016.11.041.

Author's response to references:

Thank-you for the list of references.

Please also note the supplement to this comment:
http://www.the-cryosphere-discuss.net/tc-2017-9/tc-2017-9-AC1-supplement.pdf

**Supplement:**

**A trace gas method of evaluating wind-driven air advection in the surface snow layer**

Stephen A. Drake[1], John S. Selker[2], Chad W. Higgins[2]

[revised manuscript text omitted]

15 By contrast, localized pressure changes due to wind blowing over surface features or caused by wind variability (turbulence) generate pressure changes with much higher frequency, smaller spatial extent and smaller amplitude than synoptic-scale pressure changes. Wind blowing steadily over surface features generate localized, quasi-static pressure gradients and air in snow moves in response to these wind-induced pressure gradients (Colbeck, 1989). These topographically induced pressure gradients generate quasi-stationary circulation patterns that transport gases (Massman and Frank, 2006) and form zones of

20 preferential sublimation and deposition (Albert, 2002) and therefore have a discernable advective signature. Turbulently generated pressure fluctuations induce air movement in snow (Drake et al., 2016) but the response of air contained in snow pore space to turbulent forcing above the snow or above permeable media in general is not fully understood despite considerable effort (de Lemos et al., 2006; Mößner and Radespiel, 2015; many others). Classical boundary layer theory (Beavers and Joseph, 1967) suggests that the time-averaged pressure gradient in permeable media such as snow would

25 generate Darcian flow (advection) aligned with the pressure gradient. Unlike the advection signature for topographic forcing, the turbulence signature does not exhibit quasi-static circulation patterns. Similar to turbulence, advection through a mechanically dispersive medium such as snow dissipates a concentration gradient (Scheidegger, 1954) but in this case preferentially spreads a plume more aggressively in the downstream direction.

Airflow through snow is regulated by intrinsic permeability, which is a proportionality constant in Darcy's Law and is a

30 measure of the interconnectedness of the pore space. Snow permeability is difficult to measure in field conditions but is a fundamental input parameter to model in-snow advection (Darcian flow). Currently accepted sampling techniques to obtain snow permeability include direct measurement with a flow-through permeameter and indirect measurements that infer permeability from some other measure, such as specific surface area. For example, sub-liter samples sizes are used for the typical flow-through permeameter (Courville et al., 2007 ), microtomography (Calonne et al., 2012) or an integrating sphere

Steve Drake 6/6/2017 8:56 PM

Steve Drake 6/6/2017 8:56 PM

Steve Drake 6/6/2017 8:56 PM

Steve Drake 6/6/2017 8:56 PM

Steve Drake 6/6/2017 8:56 PM

Steve Drake 6/6/2017 8:56 PM

Steve Drake 6/6/2017 8:56 PM

(Gallet et al., 2009). A near-infrared photography technique that infers SSA from reflectance (Tape et al., 2010) acquires pore space characteristics over larger areas but only in two dimensions, as do stereological measurements (Matzl and Schneebeli, 2010) for smaller sample sizes. Active acoustic techniques of inferring large-footprint, volume-averaged permeability of snow cover have shown potential (Albert, 2001; Drake et al., 2017) but these techniques are unproven for standard data collection. None of these techniques sample intrinsic permeability of large snow volumes and therefore they do not capture macroscopic changes in permeability due to snow inhomogeneities and fractures. The consequence of neglecting the variability of intrinsic permeability for modelling airflow through snow is not known.

The presence of in-snow advection has been experimentally inferred from natural convection (Sturm and Johnson, 1991) and from temperature changes caused by forced ventilation Albert and Hardy (1995), Sokratov and Sato (2000) and from $CO_2$ flux measurements (Bowling and Massman, 2011) but few measurements of natural air advection in snow have been obtained (Albert and Shultz, 2002; Huwald et al., 2012). Bulk $CO_2$ flux measurements by Massmann and Frank (2006), Seok et al. (2009), and Bowling and Massman (2011) have increased our appreciation for the role of wind-pumping in enhancing soil/snow/atmosphere exchange beyond that given by diffusion but lack the spatial and temporal granularity needed to discern between the relative roles of in-snow transport processes. A deeper understanding of the processes that link atmospheric pressure forcing to in-snow pore space response is needed if we are to accurately model how water vapor and chemically and radiatively active trace species propagate into, through, and out of the snow pore space.

The overarching goal of this experiment is to measure wind forcing above the snow and simultaneously perform high-spatial and temporal measurements of the evolution of a trace gas release in snow such that we can link wind forcing with in-snow response. Our strategy is to compare model simulations that implement a solution of the advection/diffusion equation for homogenous, permeable media with experimental measurements of dispersion of a tracer gas in snow. Anisotropy of seasonal snow has been evaluated (Calonne et al., 2012) and we do not assume snow homogeneity in our experimental design. Rather, we compare field experiments with an analytical solution for dispersion in homogenous media to highlight the influence of snow inhomogeneities. Step changes in permeability between successive snow layers further complicate the relationship between wind forcing and the in-snow advective response (Colbeck, 1991; Albert, 1996). To minimize the complicating influence of snow layering, we confined this exploration to the topmost snow layer that had been deposited by a significant snowfall event. We therefore focus this investigation on the effect that wind blowing over snow has on air movement within the topmost layer of a snowpack.

**2 Methods**

**2.1 Snow picket description**

[revised manuscript text omitted]

Steve Drake 6/6/2017 8:56 PM

Steve Drake 6/6/2017 8:56 PM
**Moved up [12]:** Location

Steve Drake 6/6/2017 8:56 PM

Steve Drake 6/6/2017 8:56 PM
**Deleted Cells**

Steve Drake 6/6/2017 8:56 PM
**Moved up [15]:** Meteorological and Surface Snow Conditions. Air Temperature; Crystal type; Size; Hardness

Steve Drake 6/6/2017 8:56 PM
**Formatted Table**

Steve Drake 6/6/2017 8:56 PM
**Deleted Cells**

Steve Drake 6/6/2017 8:56 PM

Steve Drake 6/6/2017 8:56 PM
**Moved up [16]:** Dutchman Flat, Oregon

Steve Drake 6/6/2017 8:56 PM
**Moved up [18]:** , predominantly from the SW with arrival of a surface front. Intermittent snowfall through the night.

Steve Drake 6/6/2017 8:56 PM
**Moved up [19]:** Dutchman Flat, Oregon

Steve Drake 6/6/2017 8:56 PM
**Moved up [22]:** Winds turning from SW to NW through the day.

Steve Drake 6/6/2017 8:56 PM
**Moved up [23]:** directional variability. Clearing weather through day.

Steve Drake 6/6/2017 8:56 PM
**Moved up [24]:** Santiam Pass, Oregon

Steve Drake 6/6/2017 8:56 PM
**Moved up [28]:** Storm Peak Lab, Colorado

Steve Drake 6/6/2017 8:56 PM
**Moved up [27]:** Clear day.

Steve Drake 6/6/2017 8:56 PM
**Moved up [30]:** Storm Peak Lab, Colorado

Steve Drake 6/6/2017 8:56 PM
**Moved up [33]:** Clear day.

Steve Drake 6/6/2017 8:56 PM
**Moved (insertion) [34]**

Steve Drake 6/6/2017 8:56 PM

[revised manuscript text omitted]

Steve Drake 6/6/2017 8:56 PM

Steve Drake 6/6/2017 8:56 PM

Steve Drake 6/6/2017 8:56 PM

---

## Author Comment (AC2) · 7 Jun 2017

Tuesday, June 6, 2017

Author's response to Reviewer #2 comments

Thank-you for your comments. The lead author will address your comments. Author responses reference the same page and line number as they appeared in the original manuscript. The current revision of the paper is attached as a supplement. Due to the detailed and sometimes overlapping changes requested by both reviewers we summarize all document changes as a PDF document that is appended to the manuscript.

General comments The discussion article entitled "A trace gas method of evaluating interstitial air advection and diffusion in snow" provides potentially useful data and anal-

ysis in order to address the partitioning between the various processes responsible for trace gas movement in snow. This topic is fully consistent with the scope of The Cryosphere, and requires careful experimental investigations combined with modelling approaches, which are both employed in this study. Unfortunately, the manuscript lacks precision in many respects, in particular including the description of the field experiments and numerical modelling, the terminology used, the physical properties of snow, and the literature references. In general more quantification is needed in the text, which is of- ten too vague and qualitative (although actual numbers can sometimes be found in the tables and figures, but unfortunately not used in the text).

The only general comment I have refers to the description of the processes conducive to tarce gas movements in snow. The title refers only to "advection" and "diffusion". It seems, from reading parts of the article, that the authors make a distinction between different advection processes (at least two : "quasi-static pressure gradients [...] in response to wind-induced pressure gradients"– page 1, line 19, "turbulently generated pressure fluctuations" – page 1, line 22). Given that the literature in this field is complex and there may be significant overlap between concepts used in the literature, it would be useful that this manuscript describes the processes very accurately and precisely, in order to avoid ambiguity in this central issue for this manuscript.

Besides this need for clarification, I do not have any major comment on the manuscript, having not detected any major flaw, but conversely it is often very challenging to precisely understand how the experiments (numerical and in the field) were designed, how the results were achieved, and what are their implications. I thus rather than major remarks have quite a long list of comments and points, which deserve attention from the authors in order to better convey their research to the readers. Not being myself a native english speaker, I refrain from any comments on the wording, although in several places the wording seems (perhaps) colloquial and ambiguous (I hope this can be addressed post-acceptance through the copy-editing step).

Nevertheless, I believe that a significantly improved manuscript could be accepted for

publication, given the relevance of the experiments carried out and their potential to be converted into useful scientific results.

Author's response to general comments:

The introduction and methods sections were re-written.

Specific comments

Title I think it would be appropriate to mention in the title that the emphasis is placed here on "wind-driven" processes.

Author's response to Title:

I changed the title to include "wind-driven". This change addresses a similar critique by Reviewer #1. Also, we added a description of wind-driven pressure changes to the introduction.

Abstract Page 1, line 9 : I recommend rephrasing the term "validated". It is very unclear what is meant here. "Validate" is a very strong verb.

Author's response to Page 1, line 9 : I removed the words "or validated relationships".

Page 1, line 14 : Besides "surface (chemical) reaction rates and interpretation of firn and ice cores", I think the primary implications of this work apply to interstitial water vapor movement and snow metamorphism itself, see e.g. Calonne et al. (2015). This is unfortunately almost never mentioned in this manuscript.

Author's response to Page 1, line 14 : Besides the reference to interstitial water vapor at page 1, line 21, I added references to water vapor and snow metamorphism in the introduction.

Introduction The introduction is short and does not provide a sufficiently broad status of the current knowledge in this field. For example, on the experimental work it ignores work carried out in order to measure fluxes of trace species in snow (e.g. Seok et al.,

2009). On the snow microstructure side, it only vaguelly alludes to some of the physical processes and properties involved (e.g. page 2, line 4 the word "permeability" is mentioned but the introduction lacks the establishment of the conceptual and physical framework where such variables are used). Without a proper introduction of the state of the literature and knowledge gaps, it is difficult to understand what added-value is brought by the current manuscript. I strongly recomment expanding the part from page 2, line 5 to 9 in order to better describe what previous knowledge gap is addressed by the current study, and how. At present, the introductory statement regarding the current study seems only to be a disclaimer on why only top-most snow layers are targeted, without even knowing what kind of measurements are dealt with and to serve what purpose.

Author's response to Introduction:

I rewrote the introduction to address these deficiencies.

Methods In general the "Methods" section needs considerable improvements, to enhance the clarity of the description of the experimental goals, the instruments used for CO measurements but also ancillary conditions (not only wind, but also snow properties, meteorological conditions etc.) and the experimental sites. Figure 2 is very unclear, I recommend providing pictures of the set-up in the field and conceptual sketches illustrating the actual set-up for the various configurations used (apparently implementations vary from the various cases reported). Maybe such sketches should be provided along with every result graphically shown, in order to better understand the set-up and the corresponding data. I think the first sentence of the "Methods" belongs to the Introduction (in order to better explain the goals of the manuscript), and everything else should be placed in sub-sections with informative title. The status of the part from page 2, line 10 to page 3, line 5, is unclear and often repeated with the content of sections 2.1 and 2.2 (e.g. height of the wind sensor), which is illustrative of this confusion.

Author's response to Methods:

[Figure]

I reorganized the methods section. Figure 2 was replaced with an alternate sketch of the experimental setup.

Specifically: - neutral buyoancy : The paragraph from page 2, line 10 to line 22 is quite tortuous and mixes several issues together. Line 15 it is said that "it is nearly neutrally buyoant", then line 18 "neutral buoyancy is not strictly achieved for this experiment" then line 22 recommendations are made to make the experiment "essentially neutrally buyoant". It is very hard from this text what is really at stake here, given that no quantification is provided and the text is meandering around the issue of neutral buyancy.

Author's response to neutral buoyancy: I rephrased this section.

- safety : The same paragraph states line 16 that CO is "safely handled" then line 20 it can "cause unhealthy side effects". Here again, better clarity is required.

Author's response to safety: I rephrased the section on safety.

- page 2, line 17 : reference needed here.

Author's response to page 2, line 17 : I added a reference regarding a thin film of water on ice.

- page 2, line 20 – 22 : I think such "recommendations" could be placed in the Discussion or Conclusions sections (along with a quantification of the related issues), not in the first paragraph of the Methods section.

Author's response to page 2, line 20 – 22 :

I moved this line to the conclusion as a recommendation.

- page 2, line 32 : "cases 12 through 14": at this point, the "cases" have not been introduced yet. This illustrates the need for a major reorganisation of the manuscript, in order to better streamline the description of the methods and experiments.

[Figure]
Author's response to page 2, line 32 :

I moved case information to the results section.

- Sections 2.1 and 2.2 do not seem logically organized. For example, in section 2.1, page 3, lines 13 to 20 describes the experimental approach employed to deploy the sensors and contains details relevant to the equilibration time, which could be referred to as "Deployement description", and not at all a description of the sites. Conversely, section 2.2 confuses data selection (first words of the section), quality control and operational constraints, which do not correspond a description of the deployment. This calls for a major reorganization of the Methods section.

Author's response to Sections 2.1 and 2.2: These sections were reorganized.

- section 2.1, page 3, line 9 : more information should be given on what is referred to the "broad range of wind forcing and snow permeability regimes".

Author's response to section 2.1, page 3, line 9:

I included more specific information from Table 1 in this section.

- section 2.1, page 3, line 13: what is a "low-profile snowpit" ?

Author's response to section 2.1, page 3, line 13:

I replaced "low-profile snowpit, exposing a clean face " with "shallow trench, exposing a clean face of the snow layer".

- section 2.1, page 3, line 20: what is "spongy snow" ? If this corresponds to a description of snow properties, this should be provided in the Results section, while the Methods section should describe the methods employed to characterize snow physical properties (currently missing, although this is critical for this study).

Author's response to section 2.1, page 3, line 20:

I removed the term "spongy snow" and moved this description to the results section.

Interactive
comment

Section 2.3 Page 4, line 5: that CO sensors are sensitive to humidity and temperature is already mentioned above.

Author's response to Page 4, line 5:

This information is needed to rationalize the need to calibrate the CO sensors for different snow temperatures. I don't think the repetition, briefly stated, is a problem.

Page 4, lines 9 and 10 : perhaps a sketch and pictures could be useful to better understand the functioning of the calibration chamber. At present, this is very unclear and there is a large margin for interpretation of the sentences describing the apparatus.

Author's response to Page 4, lines 9 and 10 :

I added a picture of the calibration chamber in the lab and explanation of how each calibration measurement was acquired. I also added a sketch to delineate the experimental configuration.

Page 4, line 15 and 16 : what is the definition for "cold snow" and "warm snow" calibration ? How is this implemented in practice for field measurements ?

Author's response to Page 4, line 15 and 16 :

More description of the CO sensor calibration was added to the calibration section.

Section 3 Data analysis Page 4, lines 21 to 25: these statements are not consistent with the title of the section.

Author's response to Page 4, lines 21 to 25: These lines were moved to the results section.

Page 4, line 30 : "modeled" needs to be replaced with "simulated" is what is dealt with here is actual simulation results. In addition, given that the equation used (the model) includes advection and diffusion (line 31), how is it possible that deviations between observations and simulations correspond to the influence of "advection and
snow heterogeneity" (line 30) ? This is very unclear and needs to be better described.

Author's response to Page 4, line 30 : I changed "modeled" to "simulated" and rephrased this section.

Page 4, equations (1), (2) and (3) : symbols need to be described in the text. What is D, r, t, C etc. ? What is tMAX ? Thye absence of description of the symbols hampers the understanding of the reste of the manuscript, unfortunately.

Author's response to Page 4, equations (1), (2) and (3) :

I defined the symbols in equations 1-3.

Page 4, line 10 : I con't understand whay is meant by "We calculated the diffusion coefficient for each sensor". The sensors measure the concentration of CO, I don't see how this solely can be used to compute the diffusion coefficient.

Author's response to Page 4, line 10 :

I rephrased: "We calculated the diffusion coefficient for each sensor and subtracted these values from those given by Eq. (3) to derive a residual that is an approximation of wind-driven dispersion enhancement." as: "For each CO release, we measured CO concentration as a function of time and distance from the release point to find t_MAX for each sensor. Using Eq. (3) we then calculated the diffusivity for each sensor and subtracted these values from a mono-valued diffusivity of 2.56 x 10-5 m2 s-1 consistent with snow (Huwald et al., 2012) to derive a residual that is an approximation of wind-driven dispersion enhancement."

Section 3.1 (note that, if there is only one subsection at the end of section 3, this implies that the structure is not optimal; either add more subsections from the beginning of the section 3, or drop the section 3.1 and make simple paragraph).

Author's response to Section 3.1: I reorganized Section 3 into two sections.

Page 5, line 15 ; it is surprising that a reference to Riche and Schneebeli (2012) is given

to support the fact that snow permeability could be anisotropic. First of all, anistropy of effective thermal conductivity was demonstrated by Calonne et al. (2011), but, more importantly, Calonne et al. (2012) provide direct estimates of the anisotropy of the intrinsic permeability of snow for various snow types fould in mid-latitude snow types. There is thus no need to speculate here. Given the existence of literature not accounted for in this paragraph, it probably needs full rephrasing.

Author's response to Page 5, line 15: Thank-you for your insight on this detail. I have rephrased this paragraph.

Results
 Page 5, line 23 : what is a "diffusion constant" ?

Author's response to Results
Page 5, line 23 : I changed "diffusion constant" to "diffusivity". I also replaced "diffusion coefficient" with "diffusivity" throughout the text for consistency.

Page 6, line 10 : "moderate" : rather provide numbers. "mid-to-low density" : rather provide numbers.

Author's response to Page 6, line 10 :

I replaced the descriptions with values from Table 1.

Page 7, line 6 : what is the "plume standard deviation" ? Line 7 : what do "minutely" refer to ? Earlier in the manuscript this refers to "1-minute time resolution".

Author's response to Page 7, line 6 :

I replaced "standard deviation" with "mass-weighted RMS distance from the center of mass". "Minutely" was changed to "One-minute".

Page 7, line 14: "NaN" should be defined.

Author's response to Page 7, line 14:

I replaced "NaN values" with "NaN (Not a Number) results".

[Figure]

Page 7, line 19 : is there an analytical form for Equation (1), which accounts for non-homogeneous diffusion coefficients ?

Author's response to Page 7, line 19 :

Rather than using a constant for vertical diffusivity in Equation (1), the vertical diffusivity was adjusted to decrease with depth to determine a relative influence of vertically varying diffusivity on plume evolution.

Page 7, line 23 : "higher permeability (lower density)" : better provide numbers. Furthermore, as demonstrated in Calonne et al. (2012), not only density but also specific surface area, drive variations of snow permeability.

Author's response to Page 7, line 23 :

I removed "higher permeability" and included the actual snow density from Huwald et al. (2012) and included the reference to Calonne et al. (2012). The snow density in the Huwald et al. (2012) paper was higher than for case 13.

Page 7, line 30 : "conspired" ?

Author's response to Page 7, line 30:

I changed "at which time snow heterogeneity and vertical dispersion degraded conspired to degrade" to "by which time snow heterogeneity and vertical dispersion degraded".

Page 8, line 23 : How was the diffusivity "measured" ? Up to this point in the manuscript, no apparatus measuring the diffusivity was described.

Author's response to Page 8, line 23:

I changed "measured diffusivity" to "diffusivity, ãĂŰD(tãĂŮ_MAX ), calculated from Eq. (3)".

Page 9, line 2 : what is the "smallest measured effective diffusivity" ? Over all measure-

ments across sites ? What is the value found ? How does it compare with literature values ? This is not clear.

Author's response to Page 9, line 2 :

I changed "smallest measured effective diffusivity" to "smallest calculated effective diffusivity". The value found for the calculated effective diffusivity is not relevant because it is less than the value for molecular diffusivity and therefore not physically possible.

Page 9, line 3 : note that equation (3) already exists in the manuscript (this should be equation (4)).

Author's response to Page 9, line 3 : I renumbered subsequent equations and references to equation 3.

Conclusions Page 9, line 21 : "invalidating the notion of a mono-valued diffusion coefficient over small areas" : this notion was never introduced before in the article. This statement requires that the literature review identifies the need to "invalidate" (or not) this "notion".

Author's response to Page 9, line 21 :

I both rephrased this in terms of intrinsic permeability and added a paragraph that discusses permeability to the introduction.

Page 9, line 29 : "gran-scale properties" : this is the first mention of microstructure-scale snow properties. This aspect deserves to be better introduced in the manuscript, on the basis of references suggested here as well as in the review provided by Reviewer #1. Otherwise, this manuscript will be inconsistent with current knowledge in snow physics.

Author's response to Page 9, line 29 : I rephrased this sentence.

Authors contribution: Page 10, line 10 : Given that Z. Liu and R. Hochreutner are not authors of this manuscript, I don't understand why they are mentioned here (it is of

course appropriate to mention them in the Acknowledgements).

Author's response to Page 10, line 10: I removed references to Z. Liu and R. Hochreutner from the Author contribution section.

Tables and Figures: Table 1 : More information on snow stratigraphy is needed, given that this concerns only 5 snow pits this should be doable in a condensed form. Also, in addition to mean wind-speed, its standard deviation would be useful to better assess the steadiness of the wind conditions. The content of the CO sensor depth column is not clear (and it has formatting issues, some number have a '.0", some not).

Author's response to Table 1:

I have modified the manuscript, starting with the title, to clarify that we are only concerned with the top layer of a snowpack. So stratigraphy information is not only irrelevant but also would be confusing to the reader. I added standard deviation of the wind speed to Table 1 using 20 Hz data that also slightly altered mean wind speed (relative to 1-minute averages).

Table 2 : Could be merged with Table 1. "Degree" symbol missing.

Author's response to Table 2 : I merged Table 2 with Table 1 and inserted the degree symbol.

Figure 1 : Size scale missing.

Author's response to Figure 1 : I replaced Figure 1 with an annotated picture showing CO sensor locations.

Figure 2 : Very unclear. I don't understand where is the CO cylinder and where the CO is injected into snow. The different arrangements of the "cases" could be better explained, maybe using a "3D" sketch (yet simple) which would better show how the apparatus was implemented in the field. Pictures could be used to better illustrate how the system was implemented.

Author's response to Figure 2 : Figure 2 was redesigned.

Figure 3 : The CO release position should be indicated on panels a and c

Author's response to Figure 3 : The release positions are now marked in panels a and c.

Figure 4 : I could not understand why there are several lines with the same "distance" (e.g. twice 45 cm in subplot a). Only a sketch explaing the set-up of this particular experiment could help, I'm afraid. As such it is highly ambiguous.

Author's response to Figure 4 : These issues are clarified by a revamped Figure 1.

Figure 5 : "Distances" in the labels of the axes in b, d and f should be better described (are these horizontal or vertical distances ? Where is the injection of CO (is the red star ?) ? Again, a sketch describing this experiment would be more than useful.

Author's response to Figure 5 :

Figure 1 addresses the reviewer's confusion with CO sensor spacing. Descriptions of the release point and sensor locations are also given in the figure caption. The figure caption is already long so a description of the distance is given in the text of the manuscript.

Figure 6 : "Distances" in the labels of the axes in subplot a should be better described (see comments above). Author's response to Figure 6 :

Since there is adequate space in this figure caption I added a description of the distance to the caption. A revamped Figure 2 should clarify sensor positions in this and other figures.

Figure 7 : Same comments as for Figure 5 . Furthermore, what is the red line ?

Author's response to Figure 7 : The red line is a wind barb as indicated in the figure caption.

Figure 8 : Same comments as for Figure 7. The caption refers to "impedence" which appears unique in the manuscript and is not referred to in the text.

Author's response to Figure 8 : I removed the word "impedence".

References (if not already quoted in the manuscript) Calonne, N., Flin, F., Morin, S., Lesaffre, B., du Roscoat, S. R., and Geindreau, C.: Numerical and experimental investigations of the effective thermal conductivity of snow, Geophys. Res. Lett., 38, L23501, doi: 10.1029/2011GL049234, 2011.

Calonne, N., C. Geindreau, F. Flin, S. Morin, B. Lesaffre, S. Rolland du Roscoat and P. Charrier, 2012. 3-D image-based numerical computations of snow permeabilityÂa ÌĘ: links to specific surface area, density, and microstructural anisotropy, The Cryosphere, 6, 939-951, doi: 10.5194/tc-6-939-2012, 2012.

Calonne, N., C. Geindreau, F. Flin, 2015. Macroscopic modeling of heat and water vapor transfer with phase change in dry snow based on an upscaling methodÂa ÌĘ: Influence of air convection, J. Geophys. Res.Âa ÌĘ: Earth Surf., 120, 2476-2497, doi: 10.1002/2015JF003605, 2015.

Seok, Brian, Detlev Helmig,Âa ÌĘMark Williams,Âa ÌĘDaniel Liptzin,Âa ÌĘChowanski, K.,Âa ÌĘJacques Hueber: An automated system for continuous measurements of trace gas fluxes through snow: An evaluation of the gas diffusion method at a subalpine forest site, Niwot Ridge, Colorado. Biogeochemistry, 95(1): 95-113. doi: 10.1007/s10533-009-9302-3 , 2009

Please also note the supplement to this comment:
http://www.the-cryosphere-discuss.net/tc-2017-9/tc-2017-9-AC2-supplement.pdf

[Figure]

**Supplement:**

**A trace gas method of evaluating wind-driven air advection in the surface snow layer**

[revised manuscript text omitted]

Steve Drake 6/6/2017 8:56 PM

Steve Drake 6/6/2017 8:56 PM

Steve Drake 6/6/2017 8:56 PM

Steve Drake 6/6/2017 8:56 PM

Steve Drake 6/6/2017 8:56 PM

Steve Drake 6/6/2017 8:56 PM

Steve Drake 6/6/2017 8:56 PM

(Gallet et al., 2009). A near-infrared photography technique that infers SSA from reflectance (Tape et al., 2010) acquires pore space characteristics over larger areas but only in two dimensions, as do stereological measurements (Matzl and Schneebeli, 2010) for smaller sample sizes. Active acoustic techniques of inferring large-footprint, volume-averaged permeability of snow cover have shown potential (Albert, 2001; Drake et al., 2017) but these techniques are unproven for standard data collection. None of these techniques sample intrinsic permeability of large snow volumes and therefore they do not capture macroscopic changes in permeability due to snow inhomogeneities and fractures. The consequence of neglecting the variability of intrinsic permeability for modelling airflow through snow is not known.

The presence of in-snow advection has been experimentally inferred from natural convection (Sturm and Johnson, 1991) and from temperature changes caused by forced ventilation Albert and Hardy (1995), Sokratov and Sato (2000) and from $CO_2$ flux measurements (Bowling and Massman, 2011) but few measurements of natural air advection in snow have been obtained (Albert and Shultz, 2002; Huwald et al., 2012). Bulk $CO_2$ flux measurements by Massmann and Frank (2006), Seok et al. (2009), and Bowling and Massman (2011) have increased our appreciation for the role of wind-pumping in enhancing soil/snow/atmosphere exchange beyond that given by diffusion but lack the spatial and temporal granularity needed to discern between the relative roles of in-snow transport processes. A deeper understanding of the processes that link atmospheric pressure forcing to in-snow pore space response is needed if we are to accurately model how water vapor and chemically and radiatively active trace species propagate into, through, and out of the snow pore space.

The overarching goal of this experiment is to measure wind forcing above the snow and simultaneously perform high-spatial and temporal measurements of the evolution of a trace gas release in snow such that we can link wind forcing with in-snow response. Our strategy is to compare model simulations that implement a solution of the advection/diffusion equation for homogenous, permeable media with experimental measurements of dispersion of a tracer gas in snow. Anisotropy of seasonal snow has been evaluated (Calonne et al., 2012) and we do not assume snow homogeneity in our experimental design. Rather, we compare field experiments with an analytical solution for dispersion in homogenous media to highlight the influence of snow inhomogeneities. Step changes in permeability between successive snow layers further complicate the relationship between wind forcing and the in-snow advective response (Colbeck, 1991; Albert, 1996). To minimize the complicating influence of snow layering, we confined this exploration to the topmost snow layer that had been deposited by a significant snowfall event. We therefore focus this investigation on the effect that wind blowing over snow has on air movement within the topmost layer of a snowpack.

**2 Methods**

**2.1 Snow picket description**

[revised manuscript text omitted]

25  We were not able to discriminate the relative importance of different processes that enhance in-snow air movement with a 2-D configuration of the sensor network. Nevertheless we did detect interstitial air movement suggesting a modified 3-D design that has a smaller instrument footprint than the snow pickets used in this investigation could discriminate the 3-D evolution of the tracer gas plume. Large (~ cubic meter volume), high resolution representations of permeability are not practical with current technology but in the future would enable one to discriminate between advection and changes in

30  diffusion rate due to permeability changes. Though not available in our tests, we would recommend others employ a blend of CO with standard air rather than N$_2$, which would then be essentially neutrally buoyant.

Steve Drake 6/6/2017 8:56 PM

Steve Drake 6/6/2017 8:56 PM

Steve Drake 6/6/2017 8:56 PM

Steve Drake 6/6/2017 8:56 PM

Steve Drake 6/6/2017 8:56 PM

Steve Drake 6/6/2017 8:56 PM

Steve Drake 6/6/2017 8:56 PM

Steve Drake 6/6/2017 8:56 PM

Steve Drake 6/6/2017 8:56 PM

Steve Drake 6/6/2017 8:56 PM

Steve Drake 6/6/2017 8:56 PM

Steve Drake 6/6/2017 8:56 PM

Steve Drake 6/6/2017 8:56 PM

**Acknowledgements**

We thank Dr. Noah Molotch and Dr. Michael Durand for organizing deployments at Storm Peak Lab, Colorado. We thank Lisa Dilley and USFS for arranging deployments in Deschutes National Forest, Oregon. Dr. Ziru Liu and Rebecca Hochreutener assisted with the field deployments. Data from these experiments may be obtained by corresponding with the first author.

**Author contribution**

[revised manuscript text omitted]

Steve Drake 6/6/2017 8:56 PM

Steve Drake 6/6/2017 8:56 PM
**Moved up [12]:** Location

Steve Drake 6/6/2017 8:56 PM

Steve Drake 6/6/2017 8:56 PM
**Deleted Cells**

Steve Drake 6/6/2017 8:56 PM
**Moved up [15]:** Meteorological and Surface Snow Conditions. Air Temperature; Crystal type; Size; Hardness

Steve Drake 6/6/2017 8:56 PM
**Formatted Table**

Steve Drake 6/6/2017 8:56 PM
**Deleted Cells**

Steve Drake 6/6/2017 8:56 PM

Steve Drake 6/6/2017 8:56 PM
**Moved up [16]:** Dutchman Flat, Oregon

Steve Drake 6/6/2017 8:56 PM
**Moved up [18]:** , predominantly from the SW with arrival of a surface front. Intermittent snowfall through the night.

Steve Drake 6/6/2017 8:56 PM
**Moved up [19]:** Dutchman Flat, Oregon

Steve Drake 6/6/2017 8:56 PM
**Moved up [22]:** Winds turning from SW to NW through the day.

Steve Drake 6/6/2017 8:56 PM
**Moved up [23]:** directional variability. Clearing weather through day.

Steve Drake 6/6/2017 8:56 PM
**Moved up [24]:** Santiam Pass, Oregon

Steve Drake 6/6/2017 8:56 PM
**Moved up [28]:** Storm Peak Lab, Colorado

Steve Drake 6/6/2017 8:56 PM
**Moved up [27]:** Clear day.

Steve Drake 6/6/2017 8:56 PM
**Moved up [30]:** Storm Peak Lab, Colorado

Steve Drake 6/6/2017 8:56 PM
**Moved up [33]:** Clear day.

Steve Drake 6/6/2017 8:56 PM
**Moved (insertion) [34]**

Steve Drake 6/6/2017 8:56 PM

[revised manuscript text omitted]

Steve Drake 6/6/2017 8:56 PM

Steve Drake 6/6/2017 8:56 PM

Steve Drake 6/6/2017 8:56 PM